# Gut microbiome remodeling and metabolomic profile improves in response to protein pacing with intermittent fasting versus continuous caloric restriction

The gut microbiome (GM) modulates body weight/composition and gastrointestinal functioning; therefore, approaches targeting resident gut microbes have attracted considerable interest. Intermittent fasting (IF) and protein pacing (P) regimens are effective in facilitating weight loss (WL) and enhancing body composition. However, the interrelationships between IF- and P-induced WL and the GM are unknown. The current randomized controlled study describes distinct fecal microbial and plasma metabolomic signatures between combined IF-P ($n = 21$) versus a heart-healthy, calorie-restricted (CR, $n = 20$) diet matched for overall energy intake in free-living human participants (women = 27; men = 14) with overweight/obesity for 8 weeks. Gut symptomatology improves and abundance of *Christensenellaceae* microbes and circulating cytokines and amino acid metabolites favoring fat oxidation increase with IF-P ($p < 0.05$), whereas metabolites associated with a longevity-related metabolic pathway increase with CR ($p < 0.05$). Differences indicate GM and metabolomic factors play a role in WL maintenance and body composition. This novel work provides insight into the GM and metabolomic profile of participants following an IF-P or CR diet and highlights important differences in microbial assembly associated with WL and body composition responsiveness. These data may inform future GM-focused precision nutrition recommendations using larger sample sizes of longer duration. Trial registration, March 6, 2020 (ClinicalTrials.gov as NCT04327141), based on a previous randomized intervention trial.

As a principal modulator of the gut microbiome (GM) and weight status, nutritional input holds great therapeutic promise for addressing a wide range of metabolic dysregulation[1]. Dependent on the host for nutrients and fluid, one of the main processes by which the GM affects host physiology is producing bioactive metabolites from the gastrointestinal (GI) contents. Nutrient composition, feeding frequency, and meal timing impact this dependency[2,3]. To maintain a stable community and ecosystem, the GM must regulate its growth rate and diversity in response to nutrient availability and population density[4]. Such maintenance is affected by caloric restriction (CR) coupled with periods of feeding and intermittent fasting (IF)[5]. Moreover, we've recently shown the nutritional composition and meal frequency during these periods alter the metabolizable energy for the host[6]. The current study incorporates protein pacing (P), defined as four meals/day consumed evenly spaced every 4 h, consisting of 25–50 g of protein/meal[7–9]. Indeed,

✉ e-mail: parciero@skidmore.edu

we have previously characterized a dietary approach of calorie-restricted IF-P combined and P alone[7,8]. These studies included nutrient-dense meal replacement shakes, along with whole foods, to quantitatively examine beneficial changes in body composition and cardiometabolic, inflammatory, and toxin-related outcomes in healthy and overweight individuals[7,8,10–12]. Further, recent preclinical work in mice has identified dietary protein as having anti-obesity effects after CR that are partially modulated through the GM[13]. Thus, the need to examine this in humans is warranted.

In this current work, we compare the effects of two low-calorie dietary interventions matched for weekly energy intake and expenditure; continuous caloric restriction on a heart-healthy diet (CR) aligned with current United States (US) dietary recommendations[14] versus our calorie-restricted IF-P diet[8,15], in forty-one individuals with overweight or obesity, over an 8-week intervention. We hypothesize an IF-P diet may favorably influence the GM and metabolome to a greater extent than a calorie-matched CR alone. This exploratory investigation utilizes data and samples from a randomized controlled trial (NCT04327141) that compares the effects of the CR versus IF-P diet on anthropometric and cardiometabolic outcomes, as previously published[15]. As an additional analysis, we select "high" and "low" responders based on relative weight loss (WL) for a subgroup examination of the IF-P diet to better elucidate potential differential responses to intermittent fasting and protein pacing. Of special note, one individual lost 15% of their initial body weight over the 8-week intervention; this individual is followed longitudinally for a year to explore the dynamics of their GM and fecal metabolome. Novel findings from the current study shows an IF-P regimen results in improved gut symptomatology, a more pronounced community shift, and greater divergence of the gut microbiome, including microbial families and genera, such as *Christensenellaceae*, *Rikenellaceae*, and *Marvinbryantia*, associated with favorable metabolic profiles, compared to CR. Furthermore, IF-P significantly increases cytokines linked to lipolysis, weight loss, inflammation, and immune response. These findings shed light on the differential effects of IF-P as a promising dietary intervention for obesity management and microbiotic and metabolic health.

## Results

### Intermittent fasting - protein pacing (IF-P) significantly influences gut microbiome (GM) dynamics compared to calorie restriction (CR)

We compared an IF-P vs. a CR per-protocol dietary intervention (matched for total energy intake and expenditure) over eight weeks to compare changes in weight, cardiometabolic outcomes, and the GM in men and women with overweight/obesity (IF-P: $n = 21$; CR: $n = 20$). One participant in each group were lost to follow-up due to noncompliance with dietary intervention (Fig. 1a; CONSORT flow diagram: Supplementary Fig. S1a). The primary outcomes of dietary intake, body weight and composition responses, cardiometabolic outcomes, and hunger ratings after both dietary interventions are provided in our companion paper[15]. Briefly, after a one-week run-in period consuming their usual dietary intake (baseline diet), with no differences between groups at baseline for any dietary intake variable[15], both dietary interventions significantly reduced total fat, carbohydrate, sodium, sugar, and energy intake by approximately 40% (~1000 kcals/day) from baseline levels (Fig. 1b; Supplementary Data 1). By design, IF-P increased protein intake greater than CR during the intervention. The IF-P regimen consisted of 35% carbohydrate, 30% fat, and 35% protein for five to six days per week and a weekly extended modified fasting period (36–60 h) consisting of 350–550 kcals per day using randomization, as detailed previously[7–10,15]. In comparison, the CR regimen consisted of 41% carbohydrate, 38% fat, and 21% protein in accordance with current US dietary recommendations (Supplementary Table S1)[14,16]. Using two-way factorial mixed model analysis of

variance (ANOVA), significant macronutrient decreases drove energy reduction from dietary fat and carbohydrate ($p < 0.001$), with increased protein in the IF-P compared to CR ($p < 0.001$; Supplementary Fig. S1b; Supplementary Data 1). Regarding GI functioning and GM modulation, IF-P significantly decreased sugar and increased dietary fiber relative to CR (IF-P; pre, $20 \pm 2$ vs. post, $26 \pm 2$: CR; pre, $24 \pm 3$ vs. $24 \pm 2$ g/day; $p < 0.05$). Despite similar average weekly energy intake (~9000 kcals/week) and physical activity energy expenditure (~350 kcals/day; $p = 0.260$) during the intervention, participants following the IF-P regimen lost significantly more body weight ($-8.81 \pm 0.71\%$ vs. $-5.40 \pm 0.67\%$; $p = 0.003$; Fig. 1c; Supplementary Data 1) and total, abdominal, and visceral fat mass and increased fat-free mass percentage (~2×; $p \leq 0.030$), as previously reported[15]. In addition, within-group analyses revealed a significant decrease in the reported frequency of total and lower-moderate GI symptoms (GI symptom rating score [GSRS] ≥4) over time for both IF-P and CR participants. However, when comparing the two dietary interventions at each time point, a more substantial reduction was observed in IF-P participants compared to CR participants (i.e., $-9.3\%$ vs. $-5.4\%$ and $-13.2\%$ vs. $-3.9\%$, respectively; Table 1). The increased protein and lower sugar intake in IF-P compared to CR may have favorably mediated the GM and symptomatology.

The substantial reduction in calorie intake of both groups (~40% from baseline) led us to investigate its potential impact on transient microbial colonization in the gut, as estimated by 16S rRNA gene copies (linear-mixed effects model [LME] time effect, $p = 0.114$; Fig. 1d; Supplementary Data 2). While it might be expected that a significant reduction in calorie intake could influence gut microbial colonization, our findings indicate that this reduction did not reach statistical significance within the timeframe of our study. This result contrasts with previous research that imposed more substantial energy restriction, such as a four-week regimen of ~800 kcal/day in participants with overweight/obesity, where overall gut microbial colonization notably decreased[4]. In addition to assessing microbial colonization, we also investigated whether the calorie reduction significantly influenced principal stool characteristics, including wet stool weight, Bristol stool scale (BSS), and fecal pH ($p \geq 0.066$; Table 1). However, we did not observe statistically significant changes in these parameters over the course of the study. Moreover, there were no significant differences between the two dietary intervention groups over time (interaction effect, $p \geq 0.051$). In contrast, there were significant time effects for observed amplicon sequence variants (ASVs) and phylogenetic diversity (LME time effect, $p \leq 0.023$; Fig. 1e, f; Supplementary Data 2), with values increasing at weeks four and eight compared to baseline (pairwise comparisons, $p \leq 0.048$); however, no interaction was observed for either alpha diversity metric (group × time effect, $p \geq 0.925$). To rule out the potential confounding effects of GI transit time[17], BSS (as a surrogate marker) and stool pH were not significantly correlated with alpha diversity (Spearman correlations, $p \geq 0.210$). In relation to community composition, much of the intervention variance could be attributed to individual response upon testing nested permutational analysis of variance (PERMANOVA; $R^2 = 0.749$, $p = 0.001$; Supplementary Table S2), showcasing the highly individualistic landscape of the human GM in response to dietary intervention. However, a significant 1.8% of the variance was accounted for by the group × time interaction ($p = 0.001$). Moreover, individual responses over time showed variance between the two dietary interventions (PERMANOVA, $R^2 = 0.123$, $p = 0.003$). This variability was apparent by assessing intra-individual differences, where a pronounced increase in Bray-Curtis dissimilarity was observed in the IF-P compared to the CR group after four (median Bray-Curtis dissimilarity, 0.53 [IQR: 0.47–0.61] vs. 0.38 [IQR: 0.33–0.47]) and eight weeks (0.50 [IQR: 0.41–0.55] vs. 0.39 [IQR: 0.33–0.45]; Fig. 1g; Wilcoxon rank-sum test, $p \leq 0.005$).

To understand the taxa driving this GM variation from baseline to weeks four and eight between the two dietary interventions, we

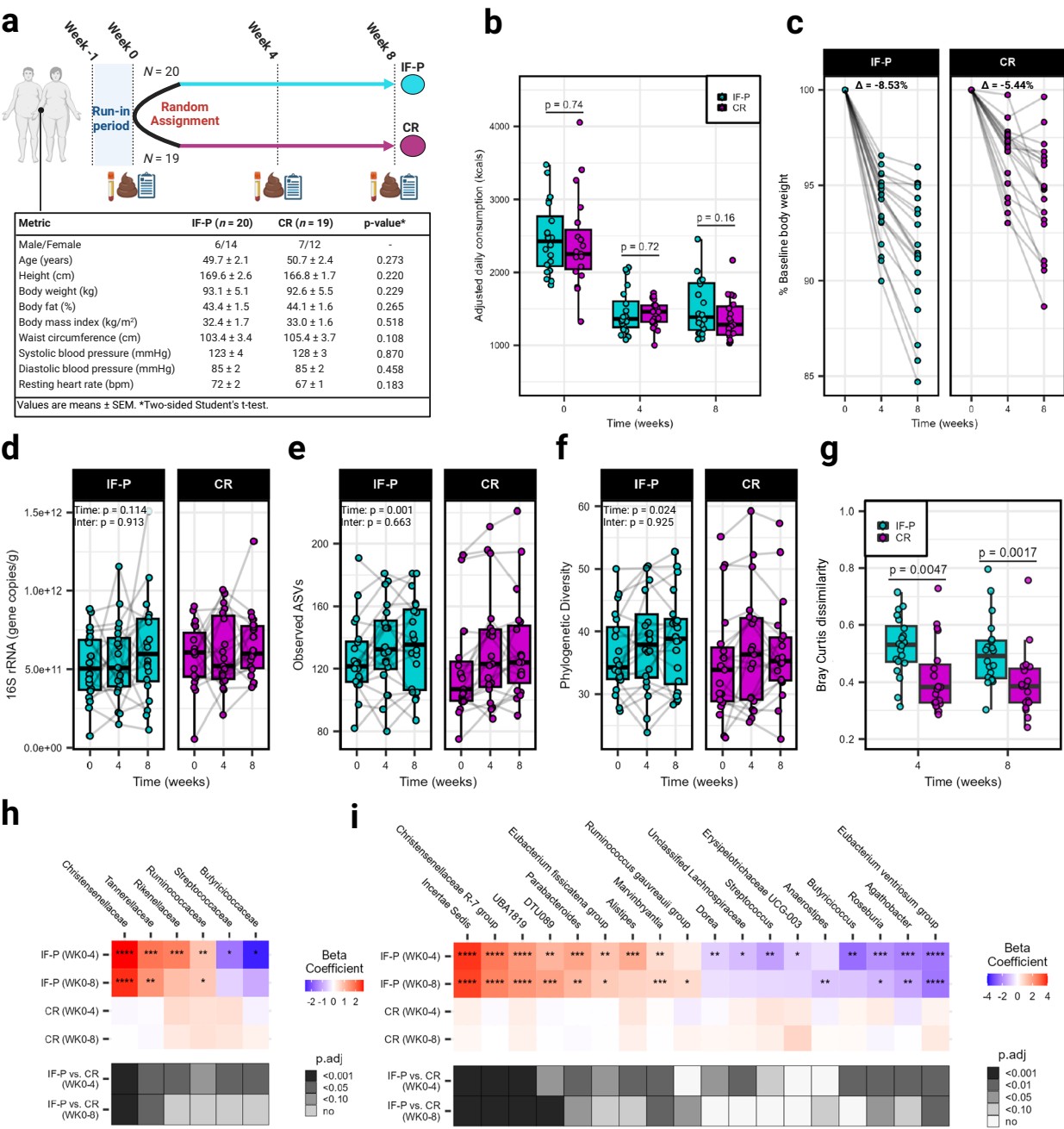

**Fig. 1 | Study characteristics and changes in the gut microbiome (GM) between intermittent fasting with protein pacing (IF-P) and continuous caloric restriction (CR) diet groups over eight weeks. a** Study design with baseline participant characteristics. A registered dietitian counseled individuals from both groups each week. Time points with data collection are shown for both IF-P and CR participants. Icons created using BioRender.com. **b** Total daily caloric intake at each time point was not significantly different between IF-P and CR diet groups (two-sided Student's *t*-test, *p* < 0.05). Adjusted values are displayed by dividing total weekly intake by seven, to account for the fasting periods of IF-P. **c** IF-P participants lost significantly more weight over time versus CR participants. Points connected by line represent percent of weight compared to baseline weight for each participant. **d** Overall gut microbial colonization, as demonstrated by qPCR-based quantification of 16S rRNA gene copies per gram wet weight was unaffected by time or intervention (linear-mixed effects [LME] model, two-sided *p* > 0.05). Alpha diversity metrics, **e** observed amplicon sequence variants (ASVs), and

**f** Phylogenetic diversity at the ASV level significantly increased over time, independent of the intervention. **g** Intra-individual changes in GM community structure from baseline to weeks four and eight in IF-P participants shifted significantly throughout the IF-P intervention compared to CR as measured by the Bray-Curtis dissimilarity index (two-sided Wilcoxon rank-sum test). All box and whiskers plots display the box ranging from the first to the third quartile, and the center the median value, while the whiskers extend from each quartile to the minimum or maximum values. Heatmap of significant changes in **h** family- and **i** genus-level bacteria by intervention. Colors indicate the within-group change beta coefficients over time for each cell, and asterisks denote significance. Black-white annotations on the bottom denote the significance of between-group change difference (by MaAsLin2 group × time interactions; *p*-values were corrected to produce adjusted values [*p*.adj] using the Benjamini–Hochberg method). For all panels, IF-P: *n* = 20, CR: *n* = 19. Source data are provided as a Source Data file.

**Table 1 | Self-reported gastrointestinal (GI) symptoms, stool characteristics, and fiber intake between intermittent fasting, protein pacing (IF-P), and continuous calorie restriction (CR) at baseline and weeks four and eight**

| Variable | Baseline | | Week 4 | | Week 8 | |
|---|---|---|---|---|---|---|
| | IF-P (n = 20) | CR (n = 19) | IF-P (n = 20) | CR (n = 19) | IF-P (n = 20) | CR (n = 19) |
| Total GI scores ≥ 2 | 130 (43.3%) | 110 (38.6%) | 102 (34%)[b] | 90 (31.6%)[a] | 76 (25.3%)[c] | 68 (23.9%)[c] |
| Total GI scores ≥ 4 | 34 (11.3%) | 27 (9.5%) | 21 (7.0%) | 18 (6.3%) | 6 (2.0%)[c] | 11 (3.9%)[b] |
| Total upper GI scores ≥ 2 | 57 (40.7%) | 45 (33.8%) | 48 (34.3%) | 40 (30.1%) | 31 (22.1%)[c] | 30 (22.6%)[a] |
| Total upper GI scores ≥ 4 | 8 (5.7%) | 13 (9.8%) | 6 (4.3%) | 6 (4.5%) | 1 (0.7%)[a] | 3 (2.3%)[a] |
| Total lower GI scores ≥ 2 | 73 (45.6%) | 65 (42.8%) | 54 (33.8%)[a] | 50 (32.9%)[a] | 45 (28.1%)[c] | 38 (25.0%)[c] |
| Total lower GI scores ≥ 4 | 26 (16.3%) | 14 (9.2%) | 15 (9.4%) | 12 (7.9%) | 5 (3.1%)[c] | 8 (5.3%) |
| Wet stool weight (g) | 90.61 ± 15.82 | 93.38 ± 17.74 | 86.16 ± 11.89 | 128.12 ± 15.45 | 112.81 ± 15.36 | 120.97 ± 19.69 |
| Bristol stool scale | 3.35 ± 0.27 | 3.84 ± 0.39 | 3.05 ± 0.30 | 3.37 ± 0.31 | 3.10 ± 0.36 | 4.00 ± 0.37 |
| Stool pH | 6.96 ± 0.13 | 6.64 ± 0.09 | 6.86 ± 0.08 | 6.68 ± 0.09 | 6.86 ± 0.09 | 6.90 ± 0.12 |
| Fiber, g/day | 19.7 ± 2.0 | 23.5 ± 3.0 | 29.3 ± 2.0[a,d] | 22.6 ± 2.0 | 26.0 ± 2.0[a,d] | 23.5 ± 2.0 |

GI scores are displayed as the sum of GI symptoms and the percent of participants reporting ≥1 symptom per category. Stool characteristic data are reported as mean ± SEM. Significant difference from baseline values as assessed by a two-sided McNemar's test: [a]$p < 0.05$; [b]$p < 0.01$; [c]$p < 0.001$; significant difference from CR: [d]$p < 0.05$. There were no significant differences between groups for GI scores at each of the time points as assessed by a two-sided Fisher's exact test ($p ≥ 0.357$) or wet stool weight, Bristol stool scale, and stool pH ($p ≥ 0.066$). Bristol stool scale evaluates the form of human feces as a marker for transit time and ranges from 1 (separate hard lumps, long transit time) to 7 (watery, short transit time). Source data are provided as a Source Data file.

constructed MaAsLin2 linear-mixed models with the individual participant as a random factor[18]. We observed differential abundance patterns at the family and genus level in response to the IF-P but not the CR intervention. Of the 28 family and 69 genus-level features captured after filtering, a respective total of six and 18 taxa displayed significant interaction effects, with all significant time effects occurring from IF-P ($p$.adj ≤ 0.10; Fig.1h, i; Supplementary Data 3, 4). Notably, the changes observed at the four-week mark were more pronounced compared to those at eight weeks. These early alterations may signify an initial adaptation phase during which microbial populations respond to the modified substrate availability and nutrient composition, suggesting a degree of community resilience[19]. Increases were sustained to the third fecal collection for the family *Christensenellaceae* and the genera *Incertae Sedis* (*Ruminococcaceae* family), *Christensenellaceae R-7 group*, and *UBA1819* (*Ruminococcaceae* family) (effect size > 2.0). *Christensenellaceae* is well regarded as a marker of a lean (anti-obesity) phenotype[20] and is associated with higher protein intake[21]. Other notable increases included *Rikenellaceae*, which, like *Christensenellaceae*, has been linked to reduced visceral adipose tissue and healthy metabolic profiles[22], and *Marvinbryantia*, a candidate marker for predicting long-term weight loss success in individuals with obesity[23]. In addition, IF-P increased *Ruminococcaceae*, which has been noted to have an increased proteolytic and lipolytic capacity[24]. This shift in IF-P participants likely represents a change in GM substrate fermentation preferences as the diet regimen (relative protein and carbohydrate) and energy restriction is expected to increase the proteolytic: saccharolytic potential ratio[25]. In contrast, all taxa that decreased in IF-P participants were butyrate producers. These included the family *Butyricicoccaceae* and several genera such as *Butyricicoccus* (week four), *Eubacterium ventriosum group* (weeks four and eight), and *Agathobacter* (week four) (effect size < −2.0). When comparing monozygotic twin pairs, *Eubacterium ventriosum group* and another reduced genus, *Roseburia*, were more abundant in the higher body mass index (BMI) siblings[26]. Others, such as the mucosa-associated *Butyricicoccus* and *Erysipelotricaceae* UCG-003, have been positively correlated with insulin resistance and speculated to contribute to impaired glycolipid metabolism[27].

Despite these changes in GM composition and increased fiber intake (+30% vs. baseline) of the IF-P participants[15], we did not detect a significant shift in the abundance of the principal fecal short-chain fatty acids (SCFAs), acetate, propionate, butyrate, or valerate, as assessed by gas chromatography-mass spectrometry (GC–MS) (LME, $p ≥ 0.470$; Supplementary Fig. S1c; Supplementary Data 5). Several factors likely contribute to this finding. For example, the distinct physical-chemical properties of fiber sources between IF-P and CR are inherently different. Participants adhering to the IF-P diet consumed most of their dietary fiber as liquid meal replacements (shakes) that are rich in non-digestible, oligosaccharide dietary-resistant starch 5 (RS5). In contrast, subjects on the CR regimen consumed their fiber from whole food sources such as vegetables, whole grains, and legumes. These fiber sources provided a mixture of soluble and insoluble fibers and a more complex fiber profile than IF-P participants. Moreover, even similar fiber profiles may function differently due to differences in food matrices and/or food preparation (cooking, raw consumption, etc.). Also of relevance is the timing of their fiber consumption. IF-P participants' fiber intake was concentrated in fiber-rich shakes, offering immediate availability of fiber to the GI tract. In contrast, CR participants consumed fiber through whole foods, leading to a slower digestion and absorption process influenced by individual digestive transit times and enzymatic profiles. Interestingly, our results parallel recent work where participants more than doubled their fiber intake without affecting fecal SCFAs[28]. The disparate findings may be due to the type of dietary-resistant starch (RS) as a component of the nutrition regimen. In the current study, RS5 was included in the meal replacement shakes (eight grams/shake, two shakes/day, 16 g/day total). Prior research supports resistant starch intakes of >20 g/day favorably modulate SCFA production, primarily butyrate, over four to 12-week interventions[29,30]. Moreover, this lack of response in fecal SCFAs in both groups may have been further compounded by the significant reduction in energy intake in both groups, where the epithelia of the GI tract may have absorbed any potential increase in SCFAs from the dietary shift. It is worth noting that stool analysis may not be the most reliable biological surrogate for capturing SCFA flux over time[28]. Nevertheless, the changes in nutrient quality, timing, ratios, and the observed shift toward proteolytic activity suggest that the luminal matrix of digesta in the IF-P group impacted substrate availability for GM. This effect appears to be an influencing force in driving the observed beneficial shifts in microbial communities, such as *Christensenellaceae* and *Incertae Sedis*, as well as improvements in GI symptomatology in IF-P compared to CR. These results underscore the complexity of dietary influences on GM and highlight the need for further research to explore the impact of liquid

meal replacements versus whole food sources on GM changes and SCFA status.

## IF-P modulates circulating cytokines and gut microbiome taxa compared to CR

Caloric restriction and WL have been well known to positively influence inflammatory cytokine expression, with GM now emerging as an important modulator[31]. Surveying a panel of 14 plasma cytokines, we noted significant interaction (group × time) effects for IL-4, IL-6, IL-8, and IL-13 (LME, $p \leq 0.034$; Fig. 2a–d; Supplementary Table S3; Supplementary Data 6). These cytokines exhibited increases at weeks four and/or eight compared to baseline exclusively in the IF-P group (pairwise comparisons, $p$.adj $\leq 0.098$), while no significant changes were observed in the CR group ($p$.adj $\geq 0.562$). Notably, IL-4 has been reported to display lipolytic effects[32], and IL-8 has been positively associated with weight loss and maintenance[33]. Regarded as a proinflammatory myokine, IL-6 can acutely increase lipid mobilization in adipose tissue under fasting or exercise conditions[34–36]. IL-13 may be important for gut mucosal immune responses and is a stimulator of mucus production from goblet cells[37], which has been recently reported to be influenced during a two-day-a-week fasting regimen in mice[38]. These results were of note considering the significant total body weight, fat, and visceral fat loss in the IF-P compared to the CR group. Surprisingly, correlational analysis with change (post – pre) in anthropometric and select plasma biomarker values with the cytokine profile did not reveal any significant associations after correcting for multiple testing effects ($p$.adj $\geq 0.476$; Supplementary Data 7). Plasma cytokines were, however, correlated with microbial composition for samples collected in the IF-P group during the intervention period (weeks four and eight) using graph-guided fused least absolute shrinkage and selection operator (GFLASSO) regression, revealing associations between cytokine-taxa pairs (Supplementary Fig. S2a). Of the four cytokines that increased in IF-P participants, we identified multiple significant correlations: *Colidextribacter* (rho = −0.55, $p$.adj = 0.015), *Ruminococcus gauvreauii group* (rho = 0.50, $p$.adj = 0.036), and *Intestinibacter* (rho = 0.45, $p$.adj = 0.086) with IL-4 (Supplementary Fig. S2b) and an unclassified genus from *Oscillospiraceae* (rho = −0.53, $p$.adj = 0.019), *Colidextribacter* (rho = −0.52, $p$.adj = 0.019), and *Ruminoccus gauvreauii group* (rho = 0.51, $p$.adj = 0.019) with IL-13 (Supplementary Fig. S2c).

Displaying negative correlations for IL-4 and IL-13, *Colidextribacter* has been shown to be positively correlated to fat accumulation, insulin, and triglyceride levels in mice fed a high-fat diet[39] and positively correlated with products of lipid peroxidation, suggesting its potential role in promoting oxidative stress[40]. Conversely, *Ruminoccus gauvreauii group* was positively correlated with IL-4 and IL-13. Although limited information is available regarding the host interactions of this microbe, this genus is considered a commensal part of the core human GM and able to convert complex polysaccharides into a variety of nutrients for their hosts[41]. While these findings highlight the potential interplay between specific microbes and cytokine profiles, the directional influence—whether microbial changes drive cytokine alterations or vice versa—cannot be determined in this study setting. Furthermore, despite the change in cytokine profiles in the IF-P group, we did not detect any significant time or group × time effects when measuring lipopolysaccharide-binding protein (LBP; Δ pre/post, IF-P: 0.24 ± 0.31 vs CR: −0.93 ± 0.49 μg/mL; $p \geq 0.254$), a surrogate marker for gut permeability[42]. While the GM plays a crucial role in modulating the gut-immune axis, the observed cytokine fluctuations and microbial associations might also involve other factors. These include the production of specific metabolites due to shifts in microbial composition as well as the influence of the dietary regimen itself, which may have a central role in shaping these interactions.

## IF-P and CR yield distinct circulating metabolite signatures and convergence of multiple metabolic pathways

To understand the potential differential impact of IF-P versus CR on the host, we surveyed the plasma metabolome, reliably detecting 136 plasma metabolites across 117 samples (i.e., QC CV < 20% and relative abundance > 1000 in 80% of samples). Based on outlier examination (random forest [RF] and principal component analysis [PCA]), no samples were categorized as outliers, and all data were retained for subsequent analysis. Metabolomic profile shifts were observed in both IF-P and CR groups compared with baseline (Canberra distance), however, these did not differ significantly by group or time (weeks four and eight; Wilcoxon rank-sum test, $p \geq 0.087$; Supplementary Fig. S3a). We prepared a general linear model (GLM) with age, sex, and time as covariates and corrected for false discovery rate (FDR). When controlling for these relevant covariates, we observed significant differences between IF-P and CR for 15 metabolites (Fig. 3a, Supplementary Table S4): 2,3-dihydroxybenzoic acid, malonic acid, choline, agmatine, protocatechuic acid, myoinositol, oxaloacetic acid, xylitol, dulcitol, asparagine, n-acetylglutamine, sorbitol, cytidine, acetylcarnitine, and urate ($p$.adj $\leq 0.089$). To estimate the univariate classification performance of the 15 significant metabolites, we performed a receiver operating characteristic (ROC) analysis. Ten metabolites demonstrated a moderate area under the curve (AUC) (0.718–0.819), while five metabolites had an AUC < 0.70. Therefore, to improve classification performance, we constructed a supervised PLS-DA model using levels of the 15 significant metabolites ($p$.adj $\leq 0.089$) and analyzed variable importance in projection (VIP) scores (Supplementary Fig. S3b). Five metabolites with a VIP > 1.0 (2,3-dihydroxybenzoic acid, malonic acid, protocatechuic acid, agmatine, and myoinositol) were retained to construct an enhanced orthogonal projection to latent structures discriminant analysis (OPLS-DA) model. In contrast, the model fit was assessed with 100-fold leave-one-out cross-validation (LOOCV; see "Methods" section). Permutation testing showed the refined OPLS-DA model to have an acceptable fit to data ($Q^2 = 0.460$, $p < 0.001$), with appreciable explanatory capacity ($R^2 = 0.506$, $p < 0.001$; Supplementary Fig. S3c). The ROC analysis produced an area under the curve (AUC) of 0.929 (95% CI: 0.868–0.973, sensitivity = 0.8, specificity = 0.9; Supplementary Fig. S3d) between the CR and IF-P groups showing good accuracy of the GLM and providing strong support for the differential expression of these 15 metabolites between groups.

Two metabolites, malonic acid, and acetylcarnitine, increased compared to the CR intervention. Several other investigators have noted the increase in acetylcarnitine via fasting protocols[43,44]. This increase is consistent with free fatty acid mobilization and increased transportation of these fatty acids via carnitine acylation into the mitochondria for fatty acid oxidation. These results would also be consistent with the expected ketogenesis, although not documented in our study, but noted by similar fasting interventions[44]. Relatedly, malonic acid, a naturally occurring organic acid, is a key regulatory molecule in fatty acid synthesis via its conversion to acetoacetate; hence, our results may reflect this increased synthesis in response to the mobilization and oxidation of fatty acids occurring during fasting. Other metabolites that decreased with IF-P include several sugar alcohols (myoinositol, dulcitol, and xylitol). Dulcitol (galactitol) is a sugar alcohol derived from galactose. It is possible that during fasting, levels of dulcitol decrease as glucose (initially) and free fatty acids (after 24–36 h of fasting) are preferentially utilized as energy substrates. One amino acid (asparagine) and one amino acid analog (N-acetylglutamine, associated with consumption of a Mediterranean diet[45]) also decreased with IF-P relative to CR. Finally, 2,3-dihydroxybenzoic acid significantly decreased with IF-P. This metabolite is formed during the metabolism of flavonoids, as it is found abundantly in fruits, vegetables, and some spices. At the cellular level, this hydroxybenzoic acid functions as a cell signaling agent and has been

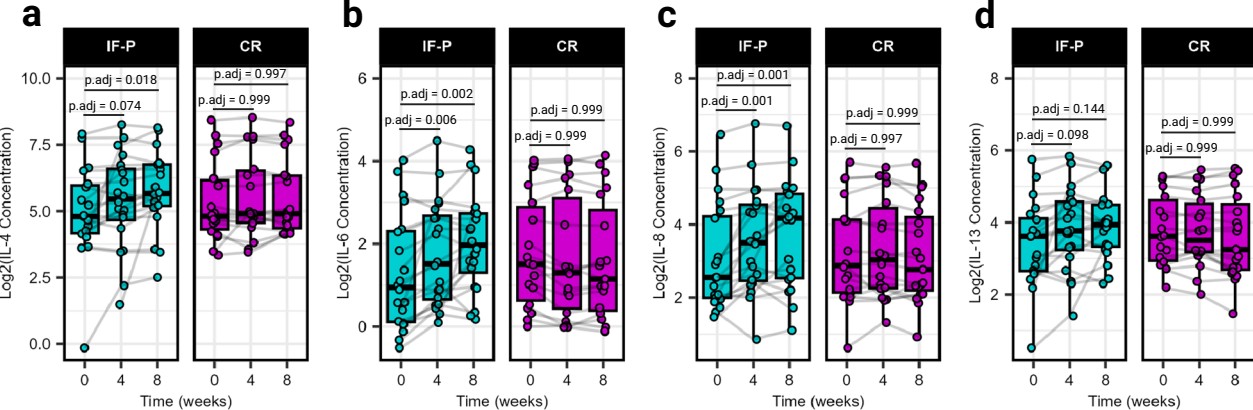

**Fig. 2 | Differences in plasma cytokine marker concentrations between the IF-P and CR diet groups. a** IL-4, **b** IL-6, **c** IL-8, and **d** IL-13: Each panel shows the cytokine concentration levels. Significant time effects and interaction effects (group × time) were detected using linear-mixed effects models (LME, two-sided $p < 0.05$), indicating differential changes over the intervention period. IF-P participants exhibited significant increases in cytokine levels compared to baseline, as evidenced by pairwise comparisons adjusted for multiple testing using the Benjamini–Hochberg method (two-sided $p$.adj < 0.10). All box and whiskers plots display the box ranging from the first to the third quartile, and the center the median value, while the whiskers extend from each quartile to the minimum or maximum values. For all panels, IF-P: $n = 20$, CR: $n = 19$. Source data are provided as a Source Data file.

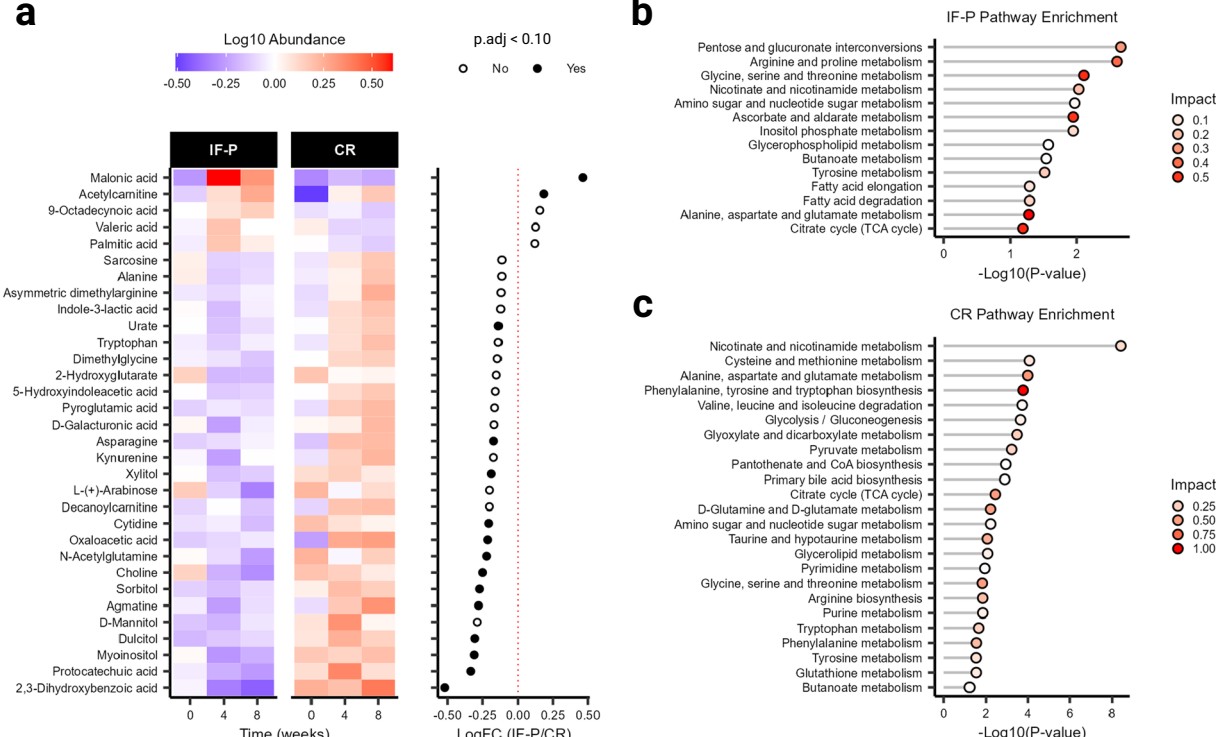

**Fig. 3 | Differences in circulating metabolite signatures and metabolic pathways between the IF-P and CR diet groups. a** Abundance and log fold-change of significant plasma metabolites between IF-P and CR groups as determined by a general linear model (GLM) adjusted for age, sex, and time. All GLM analyses utilized two-sided $p$-values, with multiple testing corrections applied using the Benjamini–Hochberg method ($p$.adj). Metabolome pathway analysis was conducted for **b** IF-P and **c** CR using all reliably detected metabolites showing significantly altered pathways ($p$.adj < 0.10) with moderate and above impact (>0.10). Impact scores were calculated using a hypergeometric test, while significance was assessed via a test of relative betweenness centrality, emphasizing the changes in metabolic network connectivity. For all panels, IF-P: $n = 20$, CR: $n = 19$. Source data are provided as a Source Data file.

speculated as a potentially protective molecule in various cancers[46]. It is unclear whether this metabolite decreased due to either dietary intake or metabolic processes related to high-protein intake or the fasting protocol. Collectively, the metabolic responses to these dietary regimens reflect the interrelationships of several anabolic and catabolic physiologic responses to three key components of these interventions: (a) the WL process itself, (b) changes in amount (and type) of

macronutrient distribution (i.e., meal replacement shakes vs. whole food diet approach; higher vs. normal protein intakes), and (c) the adherence to fasting (IF-P only).

To determine the significantly impacted pathways of the dietary interventions, we grouped participant samples according to baseline or intervention period (weeks four and eight), with IF-P and CR assessed separately. A total of 14 pathways were significant in the IF-P

group (*p*.adj < 0.10; Fig. 3b), with three displaying large impact coefficients (>0.5): (1) Glycine, serine, and threonine metabolism, (2) alanine, aspartate, and glutamate metabolism, and (3) ascorbate and aldarate metabolism. In comparison, 24 pathways were significant for the CR group (Fig. 3c), with four showing large impact coefficients (>0.5): (1) Phenylalanine, tyrosine, and tryptophan biosynthesis, (2) alanine, aspartate, and glutamate metabolism, (3) citrate cycle (TCA cycle), and (4) glycine, serine and threonine metabolism. Notably, the glycine, serine, and threonine pathway has recently been found in preclinical models to play a pivotal role in longevity and related life-sustaining mechanisms independent of diet, though heavily impacted by fasting time and caloric restriction[47]. This may be partially related to the ability of glycine to increase tissue glutathione[48,49] and protect against oxidative stress[50]. In our analysis, this pathway was significant in both diet groups and is biochemically and topologically related to the additionally captured amino acid pathway, alanine, aspartate, and glutamate metabolism, as well as the energy-releasing pathway, the citrate cycle (TCA cycle). Notably, in the CR group, phenylalanine, tyrosine, and tryptophan biosynthesis, are important for neurotransmitter production and reported to be suppressed (tryptophan) in obesity[51]. This representation may have also been attributed to the differences in protein intake[52] or differences in dietary diversity[53], yet to be determined. Regardless, we noted similar representations of pathway impact between IF-P and CR, with metabolic response centered on utilization of amino acids in addition to lipid turnover and energy pathways.

## Gut microbiome and plasma metabolome latent factors indicate differential multi-omic signatures between IF-P and CR regimens

As the plasma metabolome has been suggested as a bidirectional mediator of GM influence on the host[54], we performed a multi-omics factor analysis (MOFA)[55] to identify potential patterns of covariation and co-occurrence between the microbiome and circulating metabolites. Operating in a probabilistic Bayesian framework, MOFA simultaneously performs unsupervised matrix factorization to obtain overall sources of variability via a limited number of inferred factors and identifies shared versus exclusive variation across multiple omic data sets[55]. Eight latent factors were identified (minimum explained variance ≥2%; see "Methods" section), with the plasma metabolome and GM explaining 37.12% and 17.49% of the overall sample variability, respectively (Fig. 4a). Based on significance and the proportion of total variance explained by individual factors for each omic assay, Factors 1 ($R^2 = 11.98$) and 6 ($R^2 = 5.28$) captured the greatest covariation between the two omic layers (Fig. 4a; Supplementary Table S5). In contrast, Factors 2 and 5 were nearly exclusive to the metabolome, and factors 3 and 4 to the GM. Interestingly, Factor 1 was significantly negatively correlated to dietary protein intake (Spearman rho = −0.270, p.adj = 0.021; Fig. 4b) and captured the variation associated with the CR diet (Wilcoxon rank-sum test, *p*.adj = 3.2e-04; Fig. 4c). Factor 6 had the greatest number of significant correlations, including negative associations with visceral adipose tissue, waist circumference, body weight, BMI, fat mass, android fat, subcutaneous adipose tissue, dietary sodium, carbohydrate, fat, energy intake (kcal), and sugar (Spearman rho ≤ −0.220, *p*.adj ≤ 0.075) and captured the variation associated with IF-P (Wilcoxon rank-sum test, *p*.adj = 0.007).

Assessing the positive weights (feature importance) of Factor 1 revealed a microbial and metabolomic signature linked with CR, including the taxa *Faecalibacterium*, *Romboutsia*, and *Roseburia*, and the plasma metabolites myoinositol, agmatine, N-acetylglutamine, erythrose, and mucic acid (Fig. 4d). Previous dietary restriction studies have reported co-occurrence of gut microbial taxa and plasma metabolites that span a wide variety of applications and investigations[56]. The specific co-occurrences observed in Factor 1

exhibited an abundance of butyrate-producing bacterial taxa that utilize carbohydrates as their predominant substrate and plasma metabolites that are generally involved in carbohydrate metabolism, such as erythrose, an intermediate in the pentose phosphate pathway (PPP), and mucic acid which is derived from galactose and/or galactose-containing compounds (i.e., lactose). These co-occurrence patterns biologically cohere considering the nutritional profile of the CR group and the large contribution of fiber-rich, unrefined carbohydrates and reduction in sugar (~50% kcal from sugar). Indeed, these nutritional changes may have influenced the GM to accommodate changes in dietary substrate more efficiently. One interesting co-occurrence was the genus *Romboutsia* and metabolite N-acetylglutamine. *Romboutsia* has been shown to produce several SCFAs and ferment certain amino acids, including glutamate[57]. N-acetylglutamine is biosynthesized from glutamate; thus, its co-occurrence with the abundance of *Romboutsia* encourages further exploration into this interaction[58].

Factor 6 captured the signature associated with IF-P, with positive contributions from the taxa *Incertae Sedis* (*Ruminococcaceae* family), *Erysipelatoclostridium*, *Christensenellaceae R-7 group*, *Oscillospiraceae* UCG-002, and *Alistipes*, and the plasma metabolites malonic acid, adipic acid, succinate, methylmalonic acid, and mucic acid (Fig. 4d). Prior work has established that *Alistipes* increases from diets rich in protein and fat, and contributes to the highest number of putrefaction pathways (i.e., fermentation of undigested proteins in the GI tract) over the other commensals[59]. This could explain the co-occurrence of plasma metabolites from protein catabolism, such as 2-aminoadipid acid, adipic acid, and glutamic acid[22,59]. *Oscillospiraceae* has recently been viewed with next-generation probiotic potential, harboring positive regulatory effects in areas related to obesity and chronic inflammation[60]. Mentioned prior, recent studies have reported on the role of *Christensenellaceae* on human health, participating in host amino acid and lipid metabolism as well as fiber fermentation[20], with *Christensenellaceae R-7 group* notably evidenced to correlate with visceral adipose tissue reduction[22]. As such, the elevated abundance of microbes in the GM of IF-P participants observed in this study in tandem with the co-occurrence of metabolites indicative of protein degradation and mobilization and oxidation of fatty acids, such as methylmalonic acid, malonic acid, and succinate, presents a nascent multi-omic signature of IF-P. In addition, and more pronounced in the IF-P vs CR group, participants decreased sugar intake by ~75% (kcals) compared to baseline levels. Considering the other regimental components of IF-P, the differences in multi-omic signatures likely display the selective pressures of these two interventions.

## Gut microbiome (GM) composition is associated with weight loss (WL) responsiveness to IF-P diet

The IF-P intervention produced a microbiome and metabolomic response; however, the loss in body weight and fat across individuals varied (Fig. 5a). To provide deeper characterization and explore differential features of WL responsiveness, we performed a GM-focused subgroup analysis by employing shotgun metagenomic and untargeted fecal metabolomic surveys in 10 individuals that either achieved ≥10% loss in body weight or bordered on clinically important WL (i.e., >5% BW; herein, 'High' and 'Low' responders)[61]. Importantly, baseline characteristics between WL responder classification did not differ significantly (baseline body weight: High, 108.9 ± 30.8 vs. Low, 81.9 ± 18.1 kg, *p* = 0.117; Supplementary Table S6). Assessing the GM at the fundamental taxonomic rank, species composition showed significant separation by weight loss response evaluated by Bray-Curtis dissimilarity (group × time: $R^2 = 0.114$, p = 0.001; Fig. 5b; Supplementary Table S7), with most of the variation explained by the individual ($R^2 = 0.711$, p = 0.001). In comparison, species level alpha diversity did not differ significantly

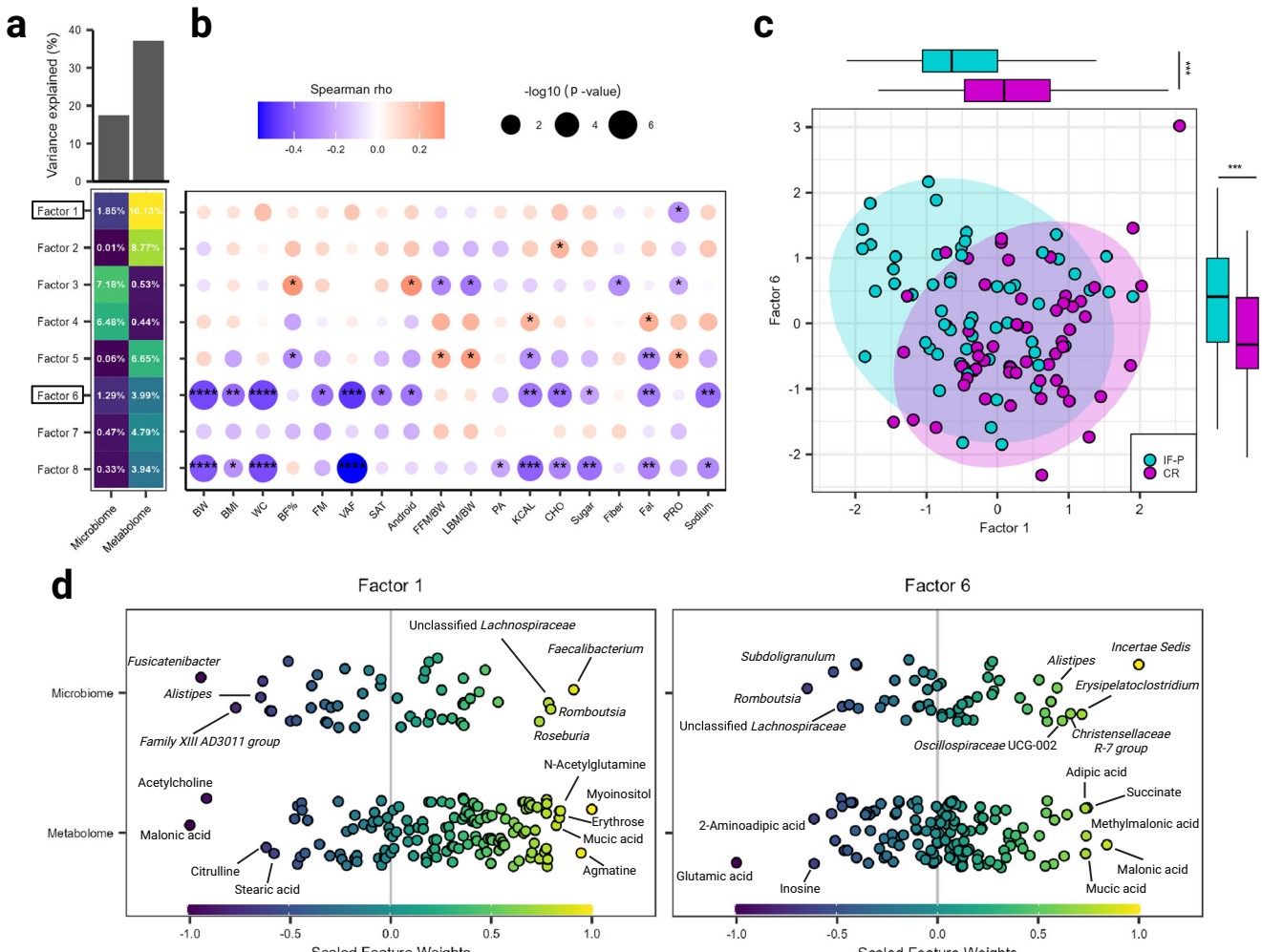

**Fig. 4 | Factors derived from the integration of the gut microbiome and plasma metabolome. a** The cumulative proportion of total variance explained ($R^2$) and proportion of total variance explained by eight individual latent factors for each omic layer. **b** Spearman correlation matrix of the eight latent factors and clinical anthropometric and dietary covariates. Each circle represents a separate association, with the size indicating the significance (-log10 (*p*-values)) and the color representing the effect size (hue) with its direction (red: positive; blue: negative). All correlations are calculated using two-sided tests. Asterisks within a circle denote significance after adjustment with the Benjamini–Hochberg method. **c** Scatter plot between classifications (group × time: $p \geq 0.674$; Fig. 5c, d). Identifying 212 species after filtering, we noted significant differences in bacterial abundances between groups over time (Fig. 5e; Supplementary Data 8). A total of 10 features increased in the High-responder group relative to the Low-response group over the eight-week study period, including *Collinsella* SGB14861, *Clostridium leptum*, *Blautia hydrogenotrophica*, and less typified species; *GGB74510 SGB47635* (unclassified Firmicutes), *GGB3511 SGB4688* (unclassified Firmicutes), *Faecalicatena contorta*, *Lachnospiraceae bacterium NSJ-29*, *Phascolarctobacterium* SGB4573, *GGB38744 SGB14842* (unclassified *Oscillospiraceae*), and *Massiliimalia timonensis* (effect size ≥ 1.163, *p*.adj ≤ 0.092). The increase in *Collinsella*, a less characterized anaerobic pathobiont that produces lactate and has been associated with low-fiber intakes[62,63] and lipid metabolism[64], may have been related to the periods of CR and IF, in conjunction with the greater influx of host-released fatty acids in the High-responder group. Relatedly, *Clostridium leptum* growth has been linked with increases in monounsaturated fat intake, reductions in blood cholesterol[65], and stimulation of Treg induction (i.e., anti-

of Factors 1 and 6, with each dot representing a sample colored by intervention. Box and whisker plots illustrate significant differences between groups after adjusting for multiple testing using the Benjamini–Hochberg method (Wilcoxon rank-sum test; top = Factor 1, *p*.adj = 3.2e-04; right = Factor 6, *p*.adj = 0.007). The plots show boxes ranging from the first to the third quartile and the median at the center, with whiskers extending to the minimum and maximum values. **d** Factor 1 and 6 loadings of genera and metabolites with the largest weights annotated. Symbols: **p*.adj < 0.10, ***p*.adj < 0.01, ****p*.adj < 0.001, *****p*.adj < 1.0e-04. For all panels, IF-P: *n* = 20, CR: *n* = 19. Source data are provided as a Source Data file.

inflammatory)[66]. The latter association is relevant to the SCFA-promoting (primarily butyrate) qualities of *Clostridium leptum*[67]. *Blautia hydrogenotrophica*, an acetogen with bidirectional metabolic cross-feeding properties (e.g., transfer of hydrogen and acetate), is also important for butyrate formation[68]. Taxa that decreased relative to the Low-responder group; *Eubacterium ventriosum*, *Streptococcus salivarius*, *Eubacterium rectale*, *Anaerostipes hadrus*, *Roseburia inulinivorans*, *Mediterraneibacter glycyrrhizinilyticus*, and *Blautia massiliensis* (effect size ≤ −1.690, *p*.adj ≤ 0.078), included butyrate producers, *Eubacterium ventriosum*, *Eubacterium rectale*, *Roseburia inulinivorans*, and others, such as *Streptococcus salivarius*, a nuclear factor kappa B (NF-κB) activity repressor[69] and Peroxisome proliferator-activated receptor gamma (PPARγ) inhibitor potentially influencing lipid and glucose metabolism[70]. Investigating monozygotic (MZ) twin pairs, *Eubacterium ventriosum* was more abundant in the higher BMI siblings[26], with enhanced scavenging fermentation capabilities[71]. *Roseburia inulinivorans* is a mobile firmicute (flagella) that harbors a wide-ranging enzymatic repertoire able to act on various dietary polysaccharide substrates suggestive

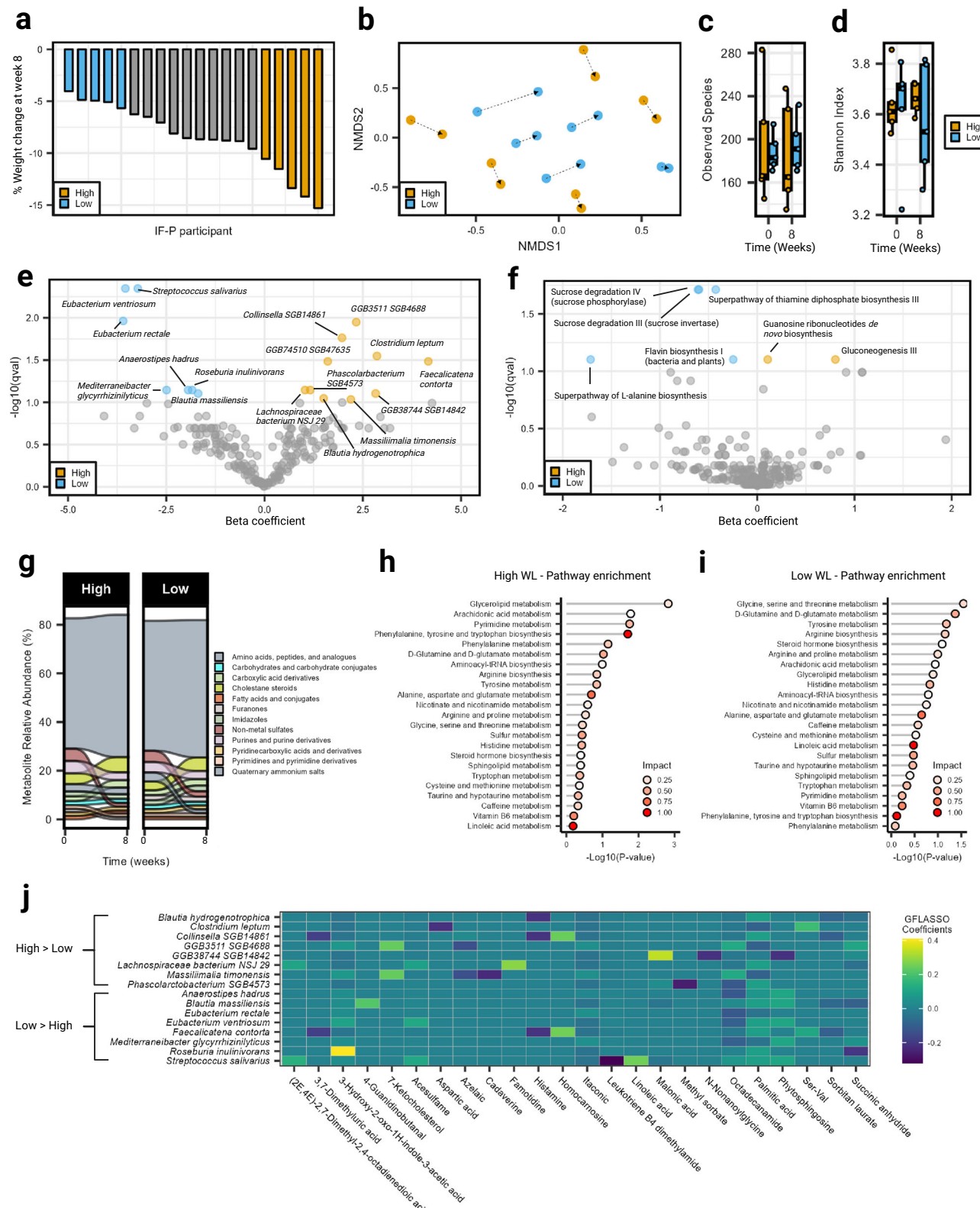

of the ability to respond to the availability of alternative dietary substrates[72]. While we noted a more variable shift in fecal total SCFAs, acetate, propionate, butyrate, or valerate (via targeted GC−MS), in the Low weight loss responders, there was no significant difference when compared to High weight loss responders (Wilcoxon rank-sum test, $p \geq 0.210$; Supplementaryl Fig. S4a; Supplementary Data 9).

Less affected compared to taxonomic features were the 275 microbial-affiliated metabolic pathways identified after filtering, of which gluconeogenesis III and guanosine ribonucleotides de novo biosynthesis were increased (effect size $\geq 0.108$, $p.adj = 0.079$), while super pathway of L-alanine biosynthesis, sucrose degradation IV (sucrose phosphorylase), sucrose degradation III (sucrose invertase), super pathway of thiamine diphosphate biosynthesis III, and

**Fig. 5 | Gut microbiome composition and metabolic differences in weight loss responsiveness to a IF-P diet. a** Relative weight loss over the eight-week intervention for each participant in the IF-P group. **b** NMDS ordination showed the personalized trajectories of participants' microbiomes over time. Dotted lines connect the same individual and point toward the final sample collection. No significant time or group × time interaction effects for alpha diversity metrics, **c** observed species, and **d** the Shannon index. Box and whiskers plots display the box ranging from the first to the third quartile, and the center the median value, while the whiskers extend from each quartile to the minimum or maximum values. Volcano plots displaying differential abundance between High and Low weight loss responders for **e** microbial species and **f** functional pathways. Significant features were more enriched in High and Low weight loss responders colored orange and light blue, respectively. **g** Alluvial plot displaying the fecal metabolite profile at the subclass level (Human Microbiome Database). Most abundant metabolite subclasses displayed (i.e., ≥1%). Metabolome pathway analysis for **h** High and **i** Low weight loss responders using all reliably detected fecal metabolites showing altered pathways with moderate and above impact (>0.10). Impact was calculated using a hypergeometric test, while significance was determined using a test of relative betweenness centrality. **j** Grid-fused least absolute shrinkage and selection operator (GFLASSO) regression of species from differential abundance analysis displayed correlative relationships with fecal metabolites. Species with greater abundance in High (High > Low) and Low (Low > High) weight loss responders are separate'. For all panels, High: $n = 5$, Low: $n = 5$. Source data are provided as a Source Data file.

flavin biosynthesis I (bacteria and plants) were decreased in the High relative to the Low weight loss responder group (effect size ≤ −0.247, $p$.adj ≤ 0.079; Fig. 5f; Supplementary Data 10)

As the difference in microbial shifts versus function is well established, we also tracked the fecal metabolome to better understand metabolic modification/production and identify potential microbial metabolic targets for future weight loss interventions. Overall, we reliably detected (QC relative standard deviation > 20% and mean intensity value > 1000 in 80% of samples) and annotated 607 (Human Metabolome Database) compounds across fecal samples. Notably, we found the fecal metabolite profile of both subgroups abundant in amino acids, peptides, and analogs, with decreases in sulfates, furanones, and quaternary ammonium salts and increases in cholestane steroids, carboxylic acid derivatives, and imidazoles (Fig. 5g). Assessing metabolite changes between groups did not yield significance when comparing logFC values (Wilcoxon rank-sum test, $p$.adj > 0.10; Supplementary Fig. S4b). Pathway analysis of High weight loss responders revealed prominent metabolic signatures relevant to lipid metabolism (glycerolipid and arachidonic metabolism), nucleotide turnover (pyrimidine metabolism), and aromatic amino acid formation (phenylalanine, tyrosine, and tryptophan biosynthesis; Fig. 5h, Supplementary Data 11). In comparison, the more prominent enriched pathways for Low weight loss responders included those related to amino acid and peptide metabolism (glycine, serine, and threonine, d-glutamine and d-glutamate, and tyrosine metabolism and arginine biosynthesis; Fig. 5i, Supplementary Data 12).

Finally, species captured by our differential abundance analysis were channeled into a GFLASSO model with the fecal metabolome library to select metabolically relevant compounds best predicted by microbial abundances. Restricting taxa and metabolites displaying stronger co-occurrence signals (GFLASSO coefficients > 0.02), we noted several patterns (Fig. 5j). This included positive associations between *GGB3511 SGB4688* (unclassified Firmicute) and malonic acid (important to fatty acid metabolism), as well as *Roseburia inulinivorans* and 3-Hydroxy-2-oxo-1H-indole-3-acetic acid. Negative associations included *Phascolarctobacterium SGB4573* with the fatty acid ester, methyl sorbate, and *Streptococcus salivarius* (anti-inflammatory) with leukotriene B4 dimethylamide.

Differences detected in our subgroup analysis suggest that the GM composition plays a role in WL responsiveness during IF-P interventions. Notable differences in taxa and fecal metabolites suggest differing substrate utilization capabilities and nutrient-acquiring pathways between High and Low responders, despite being on the same dietary regimen. Although differences between High and Low responders were statistically significant for the microbiome data, the magnitude of differences varied, suggesting further research is needed to clarify these differences.

## Long-term IF-P remodels the gut microbiome after substantial weight loss – A case study

Considering the microbiomic and metabolic importance of sustained WL, we additionally performed a longitudinal, exploratory case study analysis on the participant who lost the most body weight during the eight-week WL period (−15.3% BW, −24.9 kg). Under rigorous clinical supervision, this individual was guided through and comprehensively tracked over 52 weeks, strictly adhering to an IF-P regimen, including WL (0–16 weeks) and maintenance (16–52 weeks) periods, which included adjusting the calorie intake to maintain energy balance. Microbial richness and evenness at the species level displayed a general inverse trend with body weight reduction, although they converged at 52 weeks (Fig. 6a, b). Species dissimilarity peaked at weeks four and 16, after which it plateaued, but remained consistently higher in comparison to baseline over the 52-week period (Fig. 6c). Examining positive linear coefficients of a PERMANOVA model, constructed to detect variation between community compositions over time, dominant influences included several species within the *Lachnospiraceae* family such as *Fusicatenibacter saccharivorans*, *Blautia wexlerae*, *Blautia massillensis*, *Anaerostipes hadrus*, and *Coprococcus comes* and others like *Akkermansia muciniphila* (Fig. 6d). Negative contributions included species from the *Oscillospiraceae* family, such as *Ruminococcus bromii* and *Ruminococcus torques*. Indeed, visualizing community composition over the sampling time points suggested specific GM remodeling (Fig. 6e; Supplementary Data 13). Many keystone taxa prominent over time in the microbiome are highly relevant to the significant reduction in body weight and metabolic improvement of the case-study participant. For example, *Blautia wexlerae*, a commensal bacterium recently reported to confer anti-adipogenesis and anti-inflammatory properties to adipocytes[73] became visually more prominent over time. This association was also the case for the health-associated microbe, *Anaerostipes hadrus*, which converts inositol stereoisomers (including myoinositol) to propionate and acetate, apt to improve insulin sensitivity and reduce serum triglyceride levels[74], translating to reduced host metabolic disease risk[75]. Other elevated taxa, like the mucin-degrading *Akkermansia muciniphila* and *Bacteroides faecis*, are negatively correlated with markers for insulin resistance[76]. There was also a notable bloom of *Collinsella SGB14861* (anaerobic pathobiont producing lactate)[63] and suppression of *Eubacterium rectale*, *Ruminococcus torques* (associated with circadian rhythm disruption in mice)[77], and *Ruminococcus bromii* (an exceptional starch degrader)[78].

Compared to the more pronounced shifts in the GM, an inspection of Bray-Curtis dissimilarity at the microbial metabolic pathway level was much less affected (Supplementary Fig. S5a). Though positive contributions in multiple biosynthesis pathways were noted, as well as reductions in the superpathway of UDP-glucose-derived O-antigen building blocks biosynthesis and glucose and glucose-1-phosphate degradation (Supplementary Fig. S5b; Supplementary Data 14). We also tracked the fecal metabolome concordance with the GM to corroborate potential metabolic output. Shifts in metabolites captured by calculating the Canberra distance were prominent (Fig. 6f), with positive influences from agrocybin (possessing antifungal activity[79]), nicotinic acid (nicotinamide adenine dinucleotide precursor), and sulfate, and reductions in cadaverine (involved in the inhibition of intestinal motility[80]), maltitol, acetohydroxamic acid (a urease

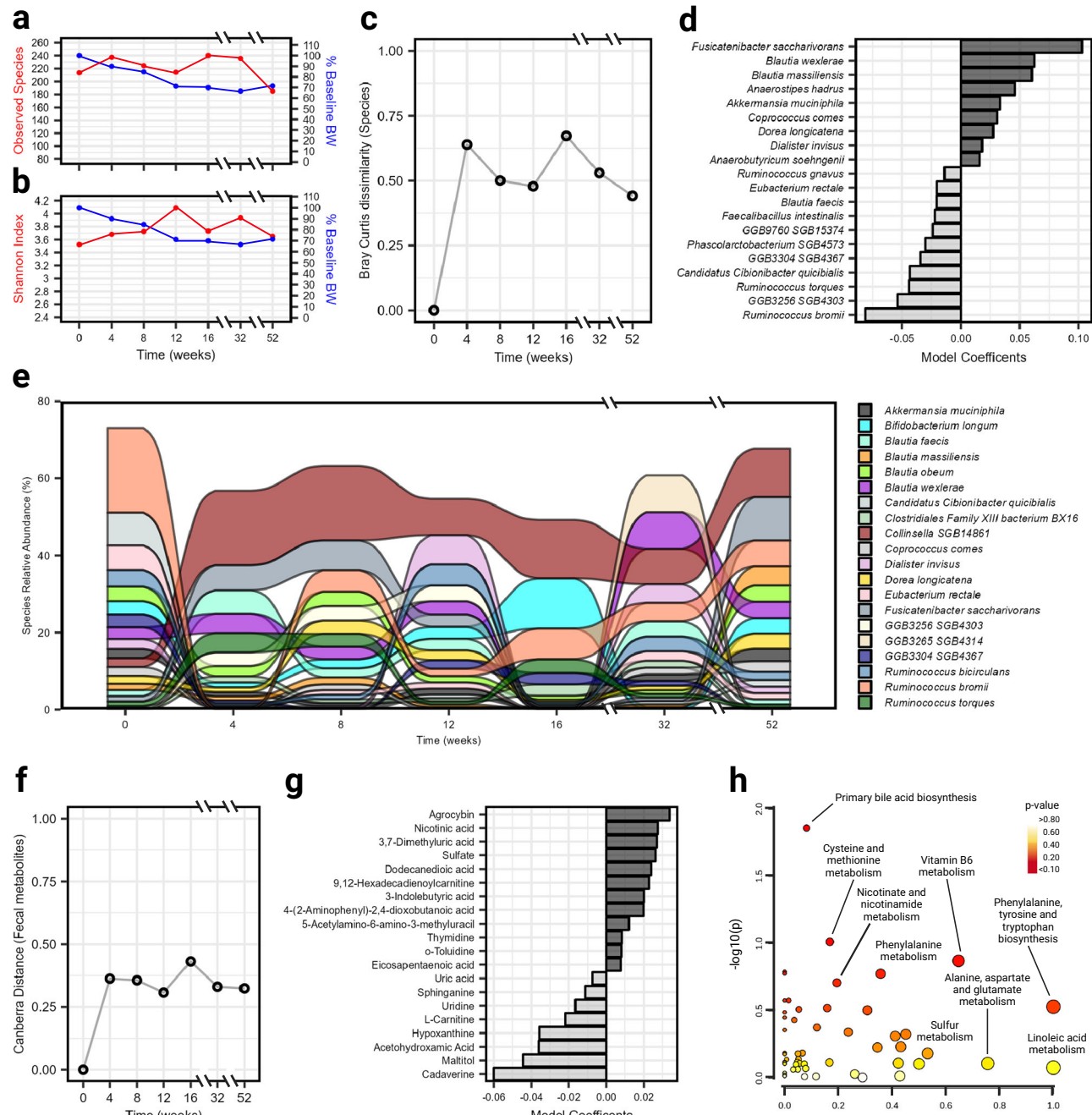

**Fig. 6 | Survey of a case-study participant's gut microbiome over a year-long period on an IF-P weight loss and maintenance regimen.** Change in alpha diversity metrics **a** observed species and **b** Shannon index with percentage of baseline body weight. **c** Bray-Curtis dissimilarity at the species level with **d** top PERMANOVA model coefficients (analysis: species-time). **e** Alluvial plot displaying the variation in abundance of the 20 most prevalent bacteria over time. For visual clarity, the less abundant taxa are not displayed. **f** Canberra distance of fecal metabolome with **g** top PERMANOVA model coefficients (analysis: pathway-time). **h** Pathway analysis of fecal metabolites comparing baseline to subsequent sample collections. Data are plotted as -log10(p) versus pathway impact. Node size corresponds to the proportion of metabolites captured in each pathway set, while node color signifies significance. Impact was calculated using a hypergeometric test, while significance was determined using a test of relative betweenness centrality. No *p*-value adjustments were made. Source data are provided as a Source Data file.

inhibitor), and hypoxanthine, after removing the dominant amino acid subclass (Fig. 6g; Supplementary Fig. S5c). At the chemical class level, we observed apparent shifts in chemical subclasses; cholestane steroids, amines, purines, and purine derivatives, and amino acids, peptides, and analogs (Supplementary Fig. S5d). Given our case-study approach, we performed a pathway analysis using all reliably detected fecal metabolites at each collection point over 52 weeks. Pathway analysis (Fig. 6h) identified primary bile acid biosynthesis ($p = 0.014$) and cysteine and methionine metabolism ($p = 0.096$) as having the

greatest significance, while the greatest impact (I) was observed in phenylalanine, tyrosine, and tryptophan biosynthesis and linoleic acid metabolism ($I = 1.0$). Alanine, aspartate, and glutamate metabolism ($I = 0.756$), vitamin B6 metabolism ($I = 0.647$), sulfur metabolism ($I = 0.532$), phenylalanine metabolism ($I = 0.357$), and nicotinate and nicotinamide metabolism ($I = 0.194$) also displayed marked pathway impacts (Supplementary Fig. S5e; Supplementary Data 15). Together, these integrated findings from the group comparisons (IF-P vs. CR), high vs. low responders, and the case study, suggest that the

remodeling of the gut microbiome through sustained weight loss on an IF-P regimen not only alters the microbial composition but also influences key metabolic pathways and output, reflective of fat mobilization and metabolic improvement.

## Discussion

Our study demonstrates distinct effects of IF-P on gut symptomatology and microbiome, as well as circulating metabolites compared to continuous CR. We observed significant changes in the GM response to both interventions; however, the IF-P group exhibited a more pronounced community shift and greater divergence from baseline (i.e., intra-individual Bray-Curtis dissimilarities). This shift was characterized by increased specific microbial families and genera, such as *Christensenellaceae*, *Rikenellaceae*, and *Marvinbryantia*, associated with favorable metabolic profiles. Furthermore, IF-P significantly increased circulating cytokine concentrations of IL-4, IL-6, IL-8, and IL-13. These cytokines have been linked to lipolysis, WL, inflammation, and immune response. The plasma metabolome analysis revealed distinct metabolite signatures in IF-P and CR groups, with the convergence of multiple metabolic pathways. These findings shed light on the differential effects of IF regimens, including IF-P as a promising dietary intervention for obesity management and microbiotic and metabolic health.

While acknowledging individual contributions of WL, protein pacing, and IF, we propose that the beneficial shifts observed may be best characterized as the culmination of features inherent in our IF-P approach. For example, it is possible that microbial competition is leveraged during reduced and intermittent nutritional input periods, emphasizing nutrient composition and food matrix type (combination of whole food and meal replacements vs. primarily whole food), affecting available substrates for gut microbes. IF-P participants' fiber intake was concentrated in fiber-rich (RS5 type) shakes, offering immediate availability of fiber to the GI tract. In contrast, CR participants consumed fiber through whole foods, leading to a slower digestion and absorption process influenced by individual digestive transit times and enzymatic profiles. This nutritional environment may create ecological niches that support symbiont microbial communities. In this investigation, we provide support of such remodeling, with intentional fasting and increased relative protein (protein pacing) consumption well-validated to improve body composition and metabolism during weight loss[7,8,15]. Our results align with previous studies on CR, where greater relative protein intake was associated with an increased abundance of *Christensenella*[81]. This increase is likely a result of increased amino acid-derived metabolites[21]. We also observed increased signatures of amino acid metabolism in the GM of IF-P participants, which may be attributed to increased nitrogen availability, prompting de novo amino acid biosynthesis. The liquid format of two of the daily meals and precise timing of high-quality protein consumption (Protein Pacing) in the IF-P regimen may have influenced these results, as amino acids play essential roles in microbial communities, acting as energy and nitrogen sources and essential nutrients for amino acid auxotrophs.

In addition to the differences in nutrient composition, the IF-P group exhibited a profound reduction (33%) in visceral fat[15]. This reduction is significant because visceral fat is highly correlated with GM. While the specific influence of GM on fat depots in our study remains unclear, the shift in cytokine profile and metabolic pathways suggests an interaction between GM and fat metabolism. Regarding GM-host interaction, we did not detect changes in gut permeability assaying LBP. However, correlations were found with cytokines IL-4 and IL-13 and microbes *Colidextribacter* (negative association) and *Ruminococcus gauveauii group* (positive association). These associations may reflect the direct impact of the dietary intervention, yet they also hint at a deeper crosstalk within the gut-immune axis. This crosstalk is known to play a pivotal role in modulating host inflammation and

influencing adipose tissue signaling pathways[42]. Furthermore, the observed microbial shifts, including changes in populations of *Christensenella*, suggest a nuanced role for certain microbes in regulating metabolic health. Notably, certain strains of *Christensenella* have been implicated in the regulation of key metabolic markers, such as glycemia and leptin levels, and in promoting hepatic fat oxidation[82].

Our findings also underscore that GM composition plays a role in WL responsiveness during IF-P interventions. Subgroup analysis based on WL responsiveness revealed significant differences in species composition at the taxonomic level. The High-responder group showed an increased abundance of certain bacteria associated with metabolic benefits and anti-inflammatory effects. In contrast, the Low-responder group exhibited an increased abundance of butyrate-producing and nutritionally adaptive species (e.g., *Eubacterium ventriosum*[71] and *Roseburia inulinivorans*[72]). Fecal metabolome analysis further highlighted differences between the two subgroups, with distinct metabolic signatures and enrichment in specific metabolic pathways. Notably, the High WL responders displayed enrichment of fecal metabolites involved in lipid metabolism. In contrast, Low responders were more prominent in pathways related to the metabolism of amino acids and peptides, including glycine, serine, and threonine, d-glutamine, and d-glutamate, as well as tyrosine metabolism and arginine biosynthesis. The latter metabolic signature has been reported in individuals with severe obesity undergoing high-protein, low-calorie diets[83]. As both High and Low WL responders were consuming the same diet, our results suggest differences in GM composition and metabolism, which could play a role in determining the success of an IF-P regimen. Though, as these enrichment analyses were performed in an exploratory manner, we acknowledge the need for a more systematic approach to validate these findings.

Finally, we provide evidence of long-term GM stabilization from these changes by following one individual over 12 months. Dietary restriction is widely used to reduce fat mass and weight in individuals with or without obesity; however, weight regain after such periods presents a critical challenge, and the underlying homeostatic mechanisms remain largely elusive. Notably, keystone taxa that became more prominent over time were associated with anti-adipogenesis, improved insulin sensitivity, and reduced metabolic disease risk. The microbial shifts were accompanied by noticeable changes in the fecal metabolome, with shifts in various metabolites and chemical subclasses. Pathway analysis identified impacts on primary bile acid biosynthesis, cysteine and methionine metabolism, and other fat mobilization and metabolic improvement pathways. These shifts were accompanied by noticeable changes in the fecal metabolome, particularly in metabolites and chemical subclasses related to lipid metabolism, nucleotide turnover, and aromatic amino acid formation.

Despite the valuable insights from our study on the complex interactions between intermittent fasting, higher protein intake using protein pacing, the GM, and circulating metabolites in obese individuals, several limitations should be acknowledged. First, our reliance on fecal samples to represent the GM may have overlooked potential microbial populations in the upper GI tract. Including samples from proximal regions in future studies would provide a more comprehensive understanding of the gut microbiome's response to IF-P and CR. In addition, the sample size for our study was determined based on the primary outcomes related to body weight and composition from the parent study[15]. This sample size may have reduced statistical power and potentially amplified individual variability among participants. However, it is important to note that the smaller RCT design allowed for more precise control over diet and lifestyle factors, minimizing potential confounding influences on the study outcomes. Furthermore, the study's duration was limited to eight weeks, which prevented potential insights into the differential long-term effects between the two interventions. However, we were able to extend the follow-up duration and conduct periodic assessments for a year in our

case-study participant, offering a more comprehensive understanding of the sustainability of the observed changes and the potential for weight regain for IF-P. The current study compared a combination of whole food and supplements (shakes and bars; IF-P) versus primarily whole food (CR), which together with variations in protein and fiber content and type may have influenced the gut symptomatology and nutrient absorption between groups. Additionally, study participants self-reported dietary intake daily, although there was close monitoring of intake through the return of empty food packaging/containers of consumed food and daily monitoring by investigators and weekly meetings with a registered dietitian. Overall, knowledge gaps are present in this research, including how the microbiome is rebuilt after food reintroduction and how overall caloric restriction and specific macronutrients contribute to this process. However, considering the multifactorial nature of weight loss and metabolic health, our work represents an important precedent for future work. Future investigators should consider integrating these factors to provide a more comprehensive understanding of the underlying mechanisms. Additional research is warranted to characterize the metabolic signature of IF-P, the time relationship between these fasting periods, and the analysis of these metabolic changes. A strength of our High-Low-responder and case-study analyses is the hypothesis-driving nature of the findings, from which targeted microbiome and/or precision nutrition interventions can be designed and tested.

In conclusion, our study provides valuable insights into the complex interactions among intermittent fasting and protein pacing, the GM, and circulating metabolites in individuals with obesity. Specifically, intermittent fasting - protein pacing significantly reduces gut symptomatology and increases gut microbes associated with a lean phenotype (*Christensenella*) and circulating cytokines mediating total body weight and fat loss. These findings highlight the importance of personalized approaches in tailoring dietary interventions for optimal weight management and metabolic health outcomes. Further research is necessary to elucidate the underlying mechanisms driving these associations and to explore the therapeutic implications for developing personalized strategies in obesity management. Additionally, future studies should consider investigating microbial populations in upper GI sections and potential intestinal tissue remodeling to gain a more comprehensive understanding of the gut microbiome's role in these interventions.

## Methods

### Study design and participants

The protocol of the clinical trial was registered on March 6, 2020 (Clinicaltrials.gov; NCT04327141), and the results of the primary analysis have been published previously[15]. Briefly, participants were recruited from Saratoga Springs, NY, and were provided informed written consent in accordance with the Skidmore College Human Subjects Institutional Review Board before participation (IRB#: 1911-859), including consent for the use of samples and data from the current study. Each procedure performed was in adherence with New York state regulations and the Federal Wide Assurance, which follows the National Commission for the Protection of Human Subjects of Biomedical and Behavioral Research, and in agreement with the Helsinki Declaration (revised in 1983). Their physicians performed a comprehensive medical examination/history assessment to rule out any current cardiovascular or metabolic disease. For at least six months before the start of the study, all eligible participants were either sedentary or lightly active (<30 min, two days/week of organized physical activity), with overweight or obesity (BMI > 27.5 kg/m2; % body fat > 30%), weight stable (±2 kg), and middle-aged (30–65 years). In addition, participants taking antibiotics, anti-fungals, or probiotics within the previous two months were excluded. Enrolled participants were matched for body weight, BMI, and body fat and randomly assigned to one of two groups: (a) IF-P

($n = 21$; 14 women; 7 men) or (b) CR ($n = 20$; 12 women; 8 men) for eight weeks. During a one-week run-in period, subjects maintained a stable body weight by consuming a similar caloric intake as their pre-enrollment caloric intake while maintaining their sedentary lifestyle. This was confirmed by matching their pre-enrollment dietary intake to the one-week run-in diet period[15]. Following baseline testing, participants were provided detailed instructions on their weight loss dietary regimen (Supplementary Table S1) and received weekly dietary counseling and compliance/adherence monitoring from the research team via daily food records, and weekly registered dietitian meetings, along with weekly visits to the Human Nutrition and Metabolism laboratory at Skidmore College (Saratoga Springs, NY) for meal distribution and empty packet/container returns. All outcome variables were assessed pre (week 0), mid (week 4), and post (week 8). All participants were compensated $100 for successful completion of the study and received an additional monthly stipend of $75 for groceries (CR group only) or up to two meals per day of food supplements and meal replacements (IF-P only).

IF days consisted of ~350–550 kcals per day, in which participants were provided a variety of supplements and snacks. Protein pacing (P) days for IF-P consisted of four and five meals/day for women and men, respectively, two of which (breakfast and one other meal) were liquid meal replacement shakes with added whole foods (Whole Blend IsaLean® Shakes, 350/400 kcals, 30/36 g of protein/meal, 9 g of fiber); a whole food evening dinner meal (450/500 kcals men), an afternoon snack (200 kcals, men only), and an evening protein snack (IsaLean® or IsaPro® Shake or IsaLean Whole Blend® Bar; 200–250 kcals). This dietary regimen provided 1350–1500 and 1700–1850 kcals/day for women and men, respectively, and a macronutrient distribution targeting 35% protein, 35% carbohydrate, 20–30 g/day of fiber, and 30% fat. Isagenix International, LLC (Gilbert, AZ, USA) provided all meal replacement shakes, bars, beverages, and supplements. In comparison, participants assigned to the CR diet followed specific guidelines of the National Cholesterol Education Program Therapeutics Lifestyle Changes (TLC) diet of the American Heart Association with a strong Mediterranean diet influence of a variety of fresh vegetables, fruits, nuts, and legumes. The specific macronutrient distribution recommended was <35% of kcal as fat; 50%–60% of kcal as carbohydrates; 15% kcal as protein; <200 mg/dL of dietary cholesterol; and 20–30 g/day of fiber. The total calorie intake was 1200 and 1500 calories per day for women and men, respectively, during the 8-week weight loss intervention. In addition to weekly meetings with the registered dietitian and daily contact with research team members, subjects were provided detailed written instructions for their meal plans. They were closely monitored through daily participant-researcher communication (e.g., email, text, and mobile phone), two-day food diary analysis, weekly dietary intake journal inspections, weekly meal/supplement container distribution, and returning empty packets and containers.

### Gastrointestinal (GI) symptom rating scale

Participants completed the 15-question GI symptom rating scale (GSRS)[84] at baseline, week four, and week eight. Briefly, each question is rated on a 7-point Likert scale (1 = absent; 2 = minor; 3 = mild; 4 = moderate; 5 = moderately severe; 6 = severe and 7 = very severe) and recalled from the previous week. Questions include symptoms related to upper abdominal pain, heartburn, regurgitation (acid reflux), empty feeling in the stomach, nausea, abdominal rumbling, bloating, belching, flatulence, and questions on defecation. The GSRS questionnaire provides explanations of each symptom, is understandable, and has reproducibility for measuring the presence of GI symptoms[85]. In our analysis, a score of ≥2 (minor) was defined as symptom presence, and a score ≥ 4 (moderate) was defined as

moderate symptom presence. Furthermore, to better categorize symptom location, bloating, flatulence, constipation, diarrhea, stool consistency, defecation urgency, and sensation of not completely emptying bowels were classified as lower GI symptoms, and nausea, heartburn, regurgitation, upper abdominal pain, empty feeling in the stomach, stomach rumbling, and belching was classified as upper GI symptoms. Total scores were also generated for overall symptom and moderate symptom presence.

## Fecal sample collection and DNA extraction

Participants were instructed to provide stool samples at baseline, week four, and week eight of the intervention. The case-study participant additionally provided samples at weeks 12, 16, 32, and 52. The entire bowel movement was collected and transported within 24 h of defecation to the Skidmore College Human Nutrition and Metabolism (Saratoga Springs, NY) laboratory using a cooler and ice packs and frozen at −80 °C. Samples were then sent to ASU (Phoenix, AZ) overnight on dry ice for analysis, where they were thawed at 4 °C and processed. Wet weight was recorded to the nearest 0.01 g after subtracting the weight of fecal collection materials. Stool samples were then rated according to the BSS[86], homogenized in a stomacher bag, and the pH was measured (Symphony SB70P, VWR International, LLC., Radnor, PA, USA). Next, the extraction of DNA was performed using the DNeasy PowerSoil Pro Kit (Cat. No. 47016, Qiagen, Germantown, MD) per the manufacturer's instructions. DNA concentration and quality were quantified using the NanoDrop™ OneC Microvolume UV-Vis Spectrophotometer (Thermo Scientific™, Waltham, MA) according to manufacturer instructions. The $OD_{260}/OD_{280}$ ratio of all samples was ≥1.80 (demonstrating DNA purity).

## Quantification of bacterial 16S rRNA genes

To estimate total bacterial biomass per sample (16S rRNA gene copies per gram of wet stool), DNA extracted from the fecal collections was assessed via quantitative polymerase chain reaction (qPCR) based on previously published methods[87,88]. Briefly, all 20 μL qPCR reactions contained 10 uL of 2X SYBR *Premix Ex Taq*™ (Tli RNase H Plus) (Takara Bio USA, Inc., San Jose, CA, USA), 0.3 μM (0.6 μL) of each primer (926 F: AAACTCAAAKGAATTGACGG; 1062 R: CTCACRRCACGAGCTGAC), 2 μL DNA template (or PCR-grade water as negative control), and 6.8 μL nuclease-free water (Thermo Fisher Scientific, Waltham, MA, USA). PCR thermal cycling conditions were as follows: 95 °C for 5 min, followed by 35 cycles of 95 °C for 15 s, 61.5 °C for 15 s, and 72 °C for 20 s, then hold at 72 °C for 5 min, along with a melt curve of 95 °C for 15 s, 60 °C for 1 min, then 95 °C for 1 s. Quantification was performed using a QuantStudio3™ Real-Time PCR System by Applied Biosystems with QuantStudio Design and Analysis Software 1.2 from Thermo Fisher Scientific (Waltham, MA, USA). All samples were analyzed in technical replicates. For quality assurance and quality control, molecular negative template controls (NTC) consisting of PCR-grade water (Invitrogen, Waltham, MA, USA) and positive controls created by linearized plasmids were run on every qPCR plate. Standard curves were run-in triplicate and used for sample quantification, ranging from $10^7$ to $10^1$ copies/μL with a cycle threshold (CT) detection limit cutoff of 33. Reaction efficiency was approximately 101%, with a slope of −3.29 and $R^2 ≥ 0.99$.

## Fecal microbiome analysis

Amplification of the 16S rRNA gene sequence was completed in triplicate PCRs using 96-well plates. Barcoded universal forward 515 F primers and 806 R reverse primers containing Illumina adapter sequences, which target the highly conserved V4 region, were used to amplify microbial DNA[89,90]. PCR, amplicon cleaning, and quantification were performed as previously outlined[90]. Equimolar ratios of amplicons from individual samples were pooled together before sequencing on the Illumina platform (Illumina MiSeq instrument,

Illumina, Inc., San Diego, CA). Raw Illumina microbial data were cleaned by removing short and long sequences, sequences with primer mismatches, uncorrectable barcodes, and ambiguous bases using the Quantitative Insights into Microbial Ecology 2 (QIIME2) software, version 2021.8[91].

16S rRNA sequencing produced 7,366,128 reads with a median of 53,776 per sample (range: 9512–470,848). Paired-end, demultiplexed data were imported and analyzed using QIIME2 software. Upon examination of sequence quality plots, base pairs were trimmed at position 20 and truncated at position 240 and were run through DADA2 to remove low-quality regions and construct a feature table using ASVs. Next, the ASV feature table was passed through the feature-classifier plugin[92], which was implemented using a naive Bayes machine-learning classifier, pre-trained to discern taxonomy mapped to the latest version of the rRNA database SILVA (138.1; 99% ASVs from 515 F/806 R region of sequences)[93]. Based on an assessment of alpha rarefaction, a threshold of 6500 sequences/sample was established, retaining all samples for downstream analysis. A phylogenic tree was then constructed using the fragment-insertion plugin with SILVA at a p-sampling depth of the rarefaction threshold to impute high-quality reads and normalize for uneven sequencing depth between samples[94]. Alpha diversity (intra-community diversity) was measured using observed ASVs and the Phylogenetic diversity index. Additionally, the Shannon index was calculated for the subgroup and case-study analyses to capture richness and evenness at the species level. Beta diversity (inter-community diversity) was measured using Bray-Curtis dissimilarity.

For shotgun metagenomics, DNA was sequenced on the Illumina NextSeq 500 platform (Illumina, CA, USA) to generate 2 × 150 bp paired-end reads at greater sequencing depth with a minimum of 10 million reads. Raw Illumina sequencing reads underwent standard quality control with FastQC. Adapters were trimmed using TrimGalore. DNA sequences were aligned to Hg38 using bowtie2[95]. DNA sequences were then analyzed via the bio bakery pipeline[96] for taxonomic composition and potential functional content with MetaPhlAn4 and HUMAnN 3.0 (UniRef90 gene-families and MetaCyc metabolic pathways), using standard parameters. Functional profiling resulted in 8528 distinct Kyoto Encyclopedia of Genes and Genomes Orthology (KO) groups and 511 metabolic pathways, which align with previous human gut microbiome studies[96].

## Blood sample collection and biochemical analyses

All participants were tested between the hours of 6:00 a.m. and 9:00 a.m., after an overnight fast for body composition assessments (height, body weight, and total body composition) at weeks 0, 4, and 8. 12-h fasted venous blood samples (~20 mL) were collected into EDTA-coated vacutainer tubes and centrifuged (Hettich Rotina 46R5) for 15 min at 4000 × g at −4 °C. After separation, plasma was stored at −80 °C until analyzed. Undiluted plasma samples were sent to Eve Technologies (Calgary, Alberta, Canada) for assessment of inflammatory cytokines [Granulocyte-macrophage colony-stimulating factor [GM-CSF], interferon-γ (IFNγ), interleukin (IL)-β, IL-2, IL-4, IL-5, IL-6, IL-8, IL-10, IL-12p70, IL-13, IL-17A, IL-23, and Tumor necrosis factor-α (TNFα)] using a high human sensitivity 14-plex cytokine assay (Millipore, Burlington, MA). Circulating LBP concentrations were quantified in duplicate using 1000x diluted plasma samples. A commercially available kit was used per the manufacturer's protocol (Cat No. EH297RB, Thermo Fisher Scientific, Inc, Waltham, MA; intra-assay coefficient variation [CV] <10%).

## Targeted plasma metabolomic analysis

For the plasma metabolomic analysis, a 12-h fasted venous blood sample (~20 mL) was collected into EDTA-coated vacutainer tubes and centrifuged (Hettich Rotina 46R5) for 15 min at 4000 × g at 4 °C. After separation, 2 mL of plasma was aliquoted and stored at −80 °C at the

Biochemistry Laboratory at Skidmore College (Saratoga Springs, NY, USA). Samples were then sent to the Arizona Metabolomics Laboratory at ASU (Phoenix, AZ, USA) overnight on dry ice for analysis, where they were thawed at 4 °C and processed. Briefly, 50 µL of plasma from each sample was processed to precipitate proteins and extract metabolites by adding 500 µL MeOH and 50 µL internal standard solution (containing 1810.5 µM $^{13}C_3$-lactate and 142 µM $^{13}C_5$-glutamic acid). The mixture was vortexed (10 s) and stored for 30 min at −20 °C, then centrifuged at 224,000 × g for 10 min at 4 °C. Supernatants (450 µL) were extracted, transferred to new Eppendorf vials, and dried (CentriVap Concentrator; Labconco, Fort Scott, KS, USA). Samples were then reconstituted in 150 µL of 40% phosphate-buffered saline (PBS)/60% acetonitrile (ACN) and centrifuged again at 22,000 × g at 4 °C for 10 min. Supernatants (100 µL) were transferred to an LC autosampler vial for subsequent analysis. Quality control (QC) was performed by creating a pooled sample from all plasma samples and injecting once every ten experimental samples to monitor system performance.

The highly-reproducible targeted LC−MS/MS method used in the current investigation was modeled after previous studies[97–99]. The specific metabolites included in our targeted detection panel are representative of more than 35 biological pathways most essential to biological metabolism and have been successfully leveraged for the sensitive and broad detection of effects related to diet[100], diseases[101], drug treatment[102], environmental contamination[103], and lifestyle factors[104]. Briefly, LC-MS/MS experiments were performed on an Agilent 1290 UPLC-6490 QQQ-MS system (Santa Clara, CA, USA). Each sample was injected twice for analysis, 10 µL using negative and 4 µL using positive ionization modes. Chromatographic separations were performed in hydrophilic interaction chromatography (HILIC) mode on a Waters Xbridge BEH Amide column (150 × 2.1 mm, 2.5 µm particle size, Waters Corporation, Milford, MA, USA). The flow rate was 0.3 mL/min, the autosampler temperature was maintained at 4 °C, and the column compartment was set at 40 °C. The mobile phase system was composed of Solvents A (10 mM ammonium acetate, 10 mM ammonium hydroxide in 95% $H_2O$/5% ACN) and B (10 mM ammonium acetate, 10 mM ammonium hydroxide in 95% ACN/5% $H_2O$). After the initial 1 min isocratic elution of 90% Solvent B, the percentage of Solvent B decreased to 40% at $t = 11$ min. The composition of Solvent B was maintained at 40% for 4 min ($t = 15$ min).

The mass spectrometer was equipped with an electrospray ionization (ESI) source. Targeted data acquisition was performed in multiple-reaction monitoring (MRM) mode. The LC−MS system was controlled by Agilent MassHunter Workstation software (Santa Clara, CA, USA), and extracted MRM peaks were integrated using Agilent MassHunter Quantitative Data Analysis software (Santa Clara, CA, USA).

## GC−MS fecal short-chain fatty acid analysis

Before GC−MS analysis of SCFAs, frozen fecal samples were first thawed overnight under 4 °C. Then, 20 mg of each sample was homogenized with 5 µL hexanoic acid−6,6,6-$d_3$ (internal standard; 200 µM in $H_2O$), 15 µL sodium hydroxide (NaOH [0.5 M]), and 500 µL MeOH. Samples were stored at −20 °C for 20 min and centrifuged at 22,000 × g for 10 min afterward. Next, 450 µL of supernatant was collected, and the sample pH was adjusted to 10 by adding 30 µL of NaOH:$H_2O$ (1:4, v-v). Samples were then dried, and the residues were initially derivatized with 40 µL of 20 mg/mL MeOX solution in pyridine under 60 °C for 90 min. Subsequently, 60 µL of MTBSTFA containing $d_{27}$-mysristic acid was added, and the mixture was incubated at 60 °C for 30 min. The samples were then vortexed for 30 s and centrifuged at 22,000 × g for 10 min. Finally, 70 µL of supernatant was collected from each sample and injected into new glass vials for GC−MS analysis.

GC−MS conditions used here were adopted from a previously published protocol[105]. Briefly, GC−MS experiments were performed on

an Agilent 7820 A GC-5977B MSD system (Santa Clara, CA); all samples were analyzed by injecting 1 µL of prepared samples. Helium was the carrier gas with a constant flow rate of 1.2 mL/min. Separation of metabolites was achieved using an Agilent HP-5 ms capillary column (30 m × 250 µm × 0.25 µm). Ramping parameters were as follows: column temperature was maintained at 60 °C for 1 min, increased at a rate of 10 °C/min to 325 °C, and then held at this temperature for 10 min. Mass spectral signals were recorded at an m/z range of 50–600, and data extraction was performed using Agilent Quantitative Analysis software. Following peak integration, metabolites were filtered for reliability. Only those with QC CV < 20% and a relative abundance of 1000 in > 80% of samples were retained for statistical analysis.

## Untargeted fecal metabolomic analysis

Briefly, each fecal sample (~20 mg) was homogenized in 200 µL MeOH:PBS (4:1, v-v, containing 1810.5 µM $^{13}C_3$-lactate and 142 µM $^{13}C_5$-glutamic Acid) in an Eppendorf tube using a Bullet Blender homogenizer (Next Advance, Averill Park, NY). Then 800 µL MeOH:PBS (4:1, v-v, containing 1810.5 µM $^{13}C_3$-lactate and 142 µM $^{13}C_5$-glutamic Acid) was added, and after vortexing for 10 s, the samples were stored at −20 °C for 30 min. The samples were then sonicated in an ice bath for 30 min. The samples were centrifuged at 22,000 × g for 10 min (4 °C), and 800 µL supernatant was transferred to a new Eppendorf tube. The samples were then dried under vacuum using a CentriVap Concentrator (Labconco, Fort Scott, KS). Prior to MS analysis, the obtained residue was reconstituted in 150 µL 40% PBS/60% ACN. A quality control (QC) sample was pooled from all the study samples.

The untargeted LC−MS metabolomics method used here was modeled after that developed and used in a growing number of studies[106–108]. Briefly, all LC−MS experiments were performed on a Thermo Vanquish UPLC-Exploris 240 Orbitrap MS instrument (Waltham, MA). Each sample was injected twice, 10 µL for analysis using negative ionization mode and 4 µL for analysis using positive ionization mode. Both chromatographic separations were performed in hydrophilic interaction chromatography (HILIC) mode on a Waters XBridge BEH Amide column (150 × 2.1 mm, 2.5 µm particle size, Waters Corporation, Milford, MA). The flow rate was 0.3 mL/min, autosampler temperature was kept at 4 °C, and the column compartment was set at 40 °C. The mobile phase was composed of Solvents A (10 mM ammonium acetate, 10 mM ammonium hydroxide in 95% $H_2O$/5% ACN) and B (10 mM ammonium acetate, 10 mM ammonium hydroxide in 95% ACN/5% $H_2O$). After the initial 1 min isocratic elution of 90% B, the percentage of Solvent B decreased to 40% at $t = 11$ min. The composition of Solvent B maintained at 40% for 4 min ($t = 15$ min), and then the percentage of B gradually went back to 90%, to prepare for the next injection. Using mass spectrometer equipped with an electrospray ionization (ESI) source, we collected untargeted data from 70 to 1050 m/z.

To identify peaks from the MS spectra, we made extensive use of the in-house chemical standards (~600 aqueous metabolites), and in addition, we searched the resulting MS spectra against the HMDB library, Lipidmap database, METLIN database, as well as commercial databases including mzCloud, Metabolika, and ChemSpider. The absolute intensity threshold for the MS data extraction was 1000, and the mass accuracy limit was set to 5 ppm. Identifications and annotations used available data for retention time (RT), exact mass (MS), MS/MS fragmentation pattern, and isotopic pattern. We used the Thermo Compound Discoverer 3.3 software for aqueous metabolomics data processing. The untargeted data were processed by the software for peak picking, alignment, and normalization. To improve rigor, only the signals/peaks with CV < 20% across quality control (QC) pools, and the signals showing up in >80% of all the samples were included for further analysis. To ensure the robustness of our model validation, we employed an enhanced validation approach by repeating the LOOCV process 100 times. Each iteration involves excluding one sample from

the dataset to serve as the test set, with the model being trained on the remaining samples. This approach, referred to as 'repeated LOOCV', was adopted to mitigate bias and provide a thorough validation of our model's predictive capability. The method signifies the number of repetitions of the LOOCV process, rather than splitting the dataset into 100 equal parts.

## Multi-omics data analysis

For MOFA, bacterial 16S rRNA ASVs and plasma metabolites were integrated using the MOFA2 package[55]. Before integration, ASV sequences were filtered (minimum of 5 ASV in greater than 10% of all samples), collapsed to the genus level, and scaled using a centralized-log-ratio, as described previously[109]. Plasma metabolites were scaled and normalized as described in the metabolome analysis. The inputs for MOFA model training comprised 53 taxa and 138 metabolites. The latent factors and feature loadings were extracted from the best-trained model with the built-in functions of MOFA2. After model fitting, the number of factors was estimated by requiring a minimum of 2% variance explained across all microbiome modalities.

Integrating microbial taxa with the same filtration as stated above (at the genus level from 16S amplicon sequencing and species level from metagenomic sequencing) and cytokine data and fecal metabolomic data, respectively, was conducted with GFLASSO (R package: GFLASSO, v0.0.0.9000). This correlation-based network solution can handle multiple response variables for a given set of predictors (in this case: 1. cytokine abundances predicted by microbial taxa response; and 2. fecal metabolite response predicted by microbial taxa). Solution parsimony was determined by an unweighted (i.e., presence or absence of association by imposing a correlation threshold) network structure. The regularization and fusion parameters were determined from the smallest root mean squared error (RMSE) estimate via cross-validation, accounting for interdependencies among microbial features. The tested parameters encompassed all combinations between $\lambda$ and $\gamma$ with values ranging from 0 to 1 (inclusive) in step increments of 0.1. GFLASSO coefficient matrices were constructed using a threshold coefficient of >0.02 to discern the strongest associative signals.

## Statistical analysis

Gastrointestinal symptom scores were on the low end of the GSRS scale and not normally distributed; therefore, nonparametric statistical tests were applied. Symptom prevalence (number of scores ≥ 2) and moderate symptom prevalence (≥4) for total, upper, and lower GI GSRS clusters were analyzed using contingency tables. Specifically, differences between IF-P and CR GI symptoms at baseline were compared using a Fisher's Exact test, whereas baseline vs. weeks four and eight values were compared with McNemar's test. Stool weight, BSS, fecal pH, plasma cytokines and LBP, and SCFAs were assessed for normality with Q-Q plots and Shapiro-Wilk tests and log-transformed where appropriate. These were then tested for time and interaction (group × time) effects using linear-mixed effect (LME) models, with each participant included as a random effect.

For analysis and visualization of the microbiome data, artifacts generated in QIIME2 were imported into the R environment (v4.2.2) using the phyloseq package (v1.42.0)[110]. Before conducting down-stream analyses, sequences were filtered to remove all non-bacterial sequences, including archaea, mitochondria, and chloroplasts. After assessing normality (Shapiro-Wilk's tests), LME models were used to test the effect of time and the interaction of group and time with the covariates of age and sex with each participant included as a random effect on the alpha diversity metrics using the nLME package (v3.1.160). For beta diversity, a nested permutational analysis of variance (PERMANOVA) was conducted on Bray-Curtis dissimilarities using the Adonis test in the vegan package (v2.6.2) with 999 permutations. The PERMANOVA model incorporated the factors of time,

individual, interaction (group × time), and participant (nested factor). A permutation test for homogeneity in multivariate dispersion (PERMDISP) was conducted using the 'betadisper' function in the vegan package to compare dispersion. To support the Adonis analysis, intra-individual differences were also compared between groups, as previously described[111], by calculating the within-subject distance for paired samples (baseline vs. weeks four and eight) and testing for group distances (Wilcoxon rank-sum test). Differential abundance analysis was performed using MaAsLin2 (v1.12.0)[18]. To detect changes in microbial features between groups over time, we built linear-mixed models that include group, time, and their interaction, with age and sex as covariates and the participant as a random factor. Before analysis, raw counts from the ASV table were filtered for any sequence not present five times in at least 30% of all samples. A significant p-value for the product term indicates that changes in microbial features differed over time between groups. The Benjamini–Hochberg (BH) procedure was used to correct for multiple testing at ≤0.10. To assess the correlation between changes in specific taxa and biomarkers over the eight-week intervention, Spearman correlation tests were performed.

Univariate and multivariate analyses of plasma metabolites and metabolic ontology analysis were performed, and results were visualized using the MetaboAnalystR 5.0[112]. Human metabolomic data were mapped to the Kyoto Encyclopedia of Genes and Genomes (KEGG) human pathway library to analyze predicted states[113]. The data were $\log_{10}$-transformed, and Pareto scaled to approximate normality before all analyses. A GLM was constructed with age, sex, and time as covariates to determine significantly affected metabolites by group intervention. Levene's test was performed to detect significant homogeneity. The BH procedure was used to correct for multiple testing at ≤0.10. Fecal metabolomic analysis for the sub-group comparison was performed by assessing logFC values between groups with a Wilcoxon rank-sum test with BH adjustment. For pathway analysis, the impact was calculated using a hypergeometric test, while significance was determined using a test of relative betweenness centrality. Importantly, the BH procedure was not applied to pathway and enzyme enrichment analyses for the sub-group assessment since these analyses involve testing the significance of multiple related hypotheses rather than independent hypotheses, which is too conservative, resulting in false negative results.

For MOFA, latent factors explaining ≥2.0% of model variance from the plasma metabolomic and amplicon microbiome data were used to perform Spearman correlations on anthropometric and nutritional data and compared between IF-P and CR groups using Wilcoxon rank-sum tests. The highest beta coefficients (>0.3) detected from GFLASSO models were further assessed by performing Spearman correlations of select microbial features with the response variables (i.e., cytokines and fecal metabolites). All statistical tests were performed with a significance level of $p < 0.05$ and BH correction of $p$.adj < 0.10. In addition, we present data in this study in accordance with the 'Strengthening The Organization and Reporting of Microbiome Studies' (STORMS) guidelines for human microbiome research[114].

## Reporting summary

Further information on research design is available in the Nature Portfolio Reporting Summary linked to this article.

# Data availability

The microbiome sequencing data generated in this study have been deposited in the BioProject Database of National Centre for Biotechnology Information database under accession code PRJNA847971. The metadata data linking the microbiome sequences with the appropriate sample ID and intervention in this study are provided in

Supplementary Data 1. The processed data are available at https://github.com/Alex-E-Mohr/GM-Remodeling-IF-ProteinPacing-vs-CaloricRestriction. Source data are provided with this paper.

## Code availability

The R code used for analysis and figure generation for reproducibility purposes are available at: https://github.com/Alex-E-Mohr/GM-Remodeling-IF-ProteinPacing-vs-CaloricRestriction.[115]

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

## Acknowledgements

We thank the trial volunteers for their dedication and commitment to the study protocol. We are grateful for the research assistants from Skidmore College who provided valuable assistance with study protocol design, scheduling, recruitment, data testing, collection, entry, and statistical analysis, and preparation of manuscripts: Molly Boyce, Jenny Zhang, Melissa Haas, Olivia Furlong, Emma Valdez, Jessica Centore, Annika Smith, Kaitlyn Judd, Aaliyah Yarde, Katy Ehnstrom, Dakembay Hoyte, Sheriden Beard, Heather Mak, and Monique Dudar. We are grateful for the extensive guidance and counseling provided by the registered dietitian Jaime Martin. We thank research coordinator Michelle Poe for her superior dedication to all aspects of the study. This study was primarily funded by an unrestricted grant from Isagenix International LLC to P.J.A. (grant #:1911-859), with secondary funding provided to K.L.S.

## Author contributions

Study conceived and designed: P.J.A. Manuscript preparation with input from all authors: A.E.M., K.L.S., D.A.B., P.J., C.M.W., D.D.S., R.K.-B., H.G., J.K.-S., K.M.A., E.G., and P.J.A. Randomized study design and execution: K.M.A., and P.J.A. Microbiome analysis: A.E.M., D.A.B., C.M.W., and R.K.-

B. Blood analyte analysis: A.E.M., K.L.S., and P.J.A. Metabolomic analysis: A.E.M., Y.J., H.G., and P.J. Statistical analysis and data presentation: A.E.M., C.M.W., D.D.S., R.K.-B., and P.J.A. Supervision and funding: K.L.S., E.G., and P.J.A.

## Competing interests

P.J.A. is a consultant for Isagenix International LLC, the study's sponsor, he is an advisory board member of the International Protein Board (iPB), and he receives financial compensation for books and keynote presentations on protein pacing (www.paularciero.com). Eric Gumpricht is employed by Isagenix International, LLC, the funding source for this research. Isagenix International, LLC had no role in the study design, data collection, analysis, or decision to publish. No authors have financial interests regarding the outcomes of this investigation. The other authors declare no competing interests.

## Additional information

**Alex E. Mohr** [1,2], **Karen L. Sweazea** [1,2,3], **Devin A. Bowes** [2], **Paniz Jasbi** [4,5], **Corrie M. Whisner** [1,2], **Dorothy D. Sears** [1], **Rosa Krajmalnik-Brown** [2], **Yan Jin** [6], **Haiwei Gu** [1,6], **Judith Klein-Seetharaman** [1,4], **Karen M. Arciero** [7], **Eric Gumpricht** [8] & **Paul J. Arciero** [7,9] ✉

[1]College of Health Solutions, Arizona State University, Phoenix, AZ, USA. [2]Biodesign Institute Center for Health Through Microbiomes, Arizona State University, Tempe, AZ, USA. [3]Center for Evolution and Medicine, College of Liberal Arts and Sciences, Arizona State University, Tempe, AZ, USA. [4]School of Molecular Sciences, Arizona State University, Tempe, AZ, USA. [5]Systems Precision Engineering and Advanced Research (SPEAR), Theriome Inc., Phoenix, AZ, USA. [6]Center of Translational Science, Florida International University, Port St. Lucie, FL, USA. [7]Human Nutrition and Metabolism Laboratory, Department of Health and Human Physiological Sciences, Skidmore College, Saratoga Springs, NY, USA. [8]Isagenix International, LLC, Gilbert, AZ, USA. [9]School of Health and Rehabilitation Sciences, Department of Sports Medicine and Nutrition, University of Pittsburgh, Pittsburgh, PA, USA. ✉e-mail: parciero@skidmore.edu

