## [Peer Review File · Nature Communications]

REVIEWER COMMENTS

Reviewer #2 (Remarks to the Author):

Review by Saar Shoer of:

“Gut microbiome remodeling and metabolomic profile in response to multimodal intermittent fasting and higher protein versus caloric restriction” by Mohr et al.

The study describes microbial and metabolomic signatures between combined intermediate fasting and protein pacing diet versus a calorie-restricted diet, in 20 obese humans over eight weeks. The paper is well written and the figures convey the main message clearly. The study includes various data modalities which is great! However, the reasoning behind using different computational methods to analyze each modality is unclear. The results should be portrayed by the magnitude of change in each diet group rather than focusing on the ones that align with the hypothesis, especially in regards to the metabolomic data.

Comments

The title is unclear, especially in “multimodal intermittent fasting and higher protein versus caloric restriction”. One can interpret it as three diets. Multimodal in what sense? For example, the description in the abstract is more coherent “between combined IF-P versus ... CR diet” or the one in figure 1 “between intermittent fasting with protein pacing (IF-P) and continuous caloric restriction (CR) diet”

Abstract, please also address the results that were in favor of the CR diet, such as the metabolites.

Abstract, as n=1 is not a sufficient sample size to draw scientific conclusions, I suggest removing it from the abstract.

Introduction, “consumed evenly spaced every 3-1/2-4 hours throughout the day”, it is unclear what these numbers refer to.

“We hypothesized an IF-P diet may favorably influence the GM and metabolome” the metabolome results are not in favor of IF-P.

“While several individuals displayed a regression to the mean response”, regression to the mean is a mathematical phenomena how can it be attributed to people?

It would be good to mention more study design details in the abstract and introduction, such as 8 weeks, obese participants, n=20, meal form - shakes vs whole foods.

To me it seems important to mention how many participants dropped out in the main text, rather just in the methods.

How was the macronutrient distribution of the IF-P diet chosen? “35% carbohydrate, 30% fat, and 110 35% protein”

Can you estimate how reliable was the monitoring? It seems challenging for an obese person to have 350 kcal per day. One way could be to check if the adherence is in association with the results.

How were the ranges determined? For example “five to six days per week” and “36-60 hours”.

Were participants blinded to the types of arms in the experiment? Did they know if they got the new proposed treatment or the control dietary regime? How could that affect the adherence and results?

When mentioning p-values, the type of test should be mentioned. Line 114, for example.

“This finding is likely due to IF-P increasing fiber intake significantly more than CR which may have favorably mediated the GM and symptomatology.” Can you support this claim? Could it be any other key components (lines 319-322) of the interventions that drove these results?

Please discuss the small number of participants, and how it may affect the results

Figure 1a is great

Figure 1b, adjusted in what way? Can you discuss the larger variance in the IF-P diet?

Figure 1d-f, text within subplots is hard to read (time, inter)

Figure 1e, please define ASV in the legend

Figure 1f, phylogenetic diversity in which phylogenetic level? species?

Figure 1h-i, can you translate the beta coefficients to comprehensible units? For example gene copies / g. No species significantly changed in the CR diet? I assume you used linear models, what were they adjusted for? Why not show the direction of change in the bottom panels as in the upper ones?

Can you hypothesize why some species changed in the 4-weeks but not in the 8-weeks? Are the comparisons 4v0 and 8v0 or 4v0 and 8v4?

Table 1 “There were no significant differences between groups for GI scores as assessed by Fisher’s exact test ($p \geq 0.357$)” at any of the time points? and for non GI scores? such as wet stool weight, Bristol stool scale and stool pH? These are likely to be affected by the eating mode

Some of these details are mentioned in the text or methods but it would be better for the figures/tables to be sufficiently informative on their own

“The substantial reduction in calorie intake of both groups (~40% from baseline) did not significantly decrease the level of transient microbial colonization in the gut” did you expect it to decrease? why? did you check whether it significantly changed? decreased or increased?

Please mention how big of a change each diet constitutes relative to the baseline of its participants, and not just between the diets. Is it the same for both diets? If it is greater in the IF-P than in the CR diet, many of the conclusions should be taken with care, such as “a pronounced increase in Bray-Curtis dissimilarity was observed in the IF-P compared to the CR group” and “We observed differential abundance patterns at the family and genus-level in response to the IF-P but not the CR intervention” as these are seemingly the expected results. Could it also be the mode of food intake, shakes vs whole foods, the reason for the differential results?

Lines 222-229, why different methods were used for the anthropometric and microbial analyses?

Explain why you focus on fiber intake throughout the paper. For example Lines 124 and 192.

“While the direction of influence wasn’t clear in this study (e.g., microbes influencing cytokine concentrations), the effect of inflammatory states is well-known to impact gut permeability.” Causality can not be determined in this study setting. The changes in cytokines and microbiome can be independent of each other, and both simply a direct result of the change in diet. In order to support the claims mentioned, one should conduct a mediation analysis to show there is an indirect effect between the two factors.

Bray-Curtis distance was used for the microbiome, while Canberra distance was used for metabolites, why?

“100-fold leave-one-out cross-validation” please explain how you ended up with 100 samples at this stage.

Lines 270-290, the motivation behind this multi-step analysis is unclear, why do the metabolites require a different approach than previously mentioned modalities (such as microbial families or cytokines)?

“To determine the significantly impacted pathways of the dietary interventions, we grouped participant samples according to baseline or intervention period (weeks 4 and 8), with IF-P and CR assessed separately.” Again, why is this analysis conducted in a different way than previous ones?

A description of the CR metabolome results is missing

Both sections of the results regarding metabolites that are displayed in Figure 3, show greater change in the CR than in the IF-P diet. However, the text is focused on the IF-P group. The authors try to mitigate this point by saying “the greater number of affected pathways and metabolites in the CR group may have been related to the overall greater dietary diversity”. To the reader, it is unclear there is a dietary diversity difference between the groups, and in general this claim does not settle with the microbiome results that were more abundant in the IF-P group

“the greater number of affected pathways and metabolites in the CR group may have been related to the overall greater dietary diversity” Can you support this claim? Could it be any other key components (lines 319-322) of the interventions that drove these results?

“we performed a multi-omics factor analysis (MOFA) to identify potential patterns of covariation and co-occurrence between the microbiome and circulating metabolites” Again, why is this analysis conducted in a different way than previous ones?

Figure 5 c-d, discuss the large variance in the high and then low groups, respectively

Authors say “Assessing metabolite changes between groups did not yield significance when comparing logFC values” meaning no change, and “Pathway analysis of High weight loss responders revealed...” statistical test or significance is not mentioned, but then “Differences detected in our subgroup analysis suggest that the GM composition plays a role in WL responsiveness during IF-P interventions. Notable differences in taxa and fecal metabolites suggest differing substrate utilization capabilities and nutrient-acquiring pathways between High and Low responders, despite being on the same dietary regimen. Although differences between High and Low responders were statistically significant, the magnitude of differences varied, suggesting further research is needed to clarify these differences.” unless I am missing something, the conclusions are not supported by the results

“Examining positive linear coefficients of a PERMANOVA model, constructed to detect variation between community compositions over time” Again, why use a different method than in the previous species analysis?

Please use scientific statistical language and methods, line 541 “visually”, line 547 “notable”, line 574 “observed”. It is not clear why these results were described without scientific testing, while the results from line 576 onwards, were analyzed by a statistical method

Line 562, until now, and again in line 568, Canberra distance was used for metabolic distance, why here it is Bray-Curtis?

“after removing the dominant amino acid subclass” why in the analysis displayed in Figure 5g you did not do the same?

Lines 584-587, as this section includes a sample size of 1 please refrain from general conclusions

“the IF-P group exhibited a more pronounced community shift and greater divergence from baseline”, this is true for some measurements but not for others such as the metabolites. Figure 3 b-c

Again, results say “Assessing metabolite changes between groups did not yield significance when comparing logFC values” so how come the conclusion is “Fecal metabolome analysis further highlighted differences between the two subgroups”

“Notably, the High WL responders displayed enrichment of fecal metabolites involved in lipid metabolism. In contrast, Low responders were more prominent in pathways related to the metabolism of amino acids and peptides, including glycine, serine, and threonine, d-glutamine and d-glutamate, as well as tyrosine metabolism and arginine biosynthesis. The latter metabolic signature has been reported in individuals with severe obesity undergoing high-protein, low-calorie diets.” as commented before, as readers we did not see any scientific analysis to support that

The limitation paragraph in the discussion is good, please add other limitations that came up in this review such as the food intake mode (shakes vs solid food) that could affect among other things gut symptomatology, and the magnitude of dietary nutrients change which could affect the magnitude of change seen in each group. Another limitation is that dietary intake and adherence are self reported

“These findings highlight the importance of personalized approaches” can you explain how this study relates to personalization? the connection is unclear

Out of curiosity, have you tried comparing the 16S species composition results to those obtained from MetaPhlan

I do not have the wet-lab expertise to review the experimental methods, but they seem very detailed. The computational methods on the other hand should be more elaborated

“For MOFA, bacterial 16S rRNA ASVs and plasma metabolites were integrated using the 941 MOFA2 package ... Integrating microbial taxa ... cytokine data and fecal metabolomic data, respectively, was conducted with GFLASSO”, please explain the motivation for using different methods

“After assessing normality (Shapiro-Wilk’s tests), LME models were used to test the effect of time and the interaction of group and time with each participant included as a random effect on the alpha diversity metrics using the nLME package. For beta diversity, a nested permutational analysis of variance (PERMANOVA) was conducted on Bray-Curtis dissimilarities using the Adonis test in the vegan package (v2.6.2) with permutations.” why use different methods?

I would split the computational methods description, as right now, if someone is looking for details of a specific analysis, one need to read the entire section

“To detect changes in microbial features between groups over time, we built linear mixed models” and “Univariate and multivariate analyses of plasma metabolites and metabolic ontology analysis were performed” why use different methods?

“The Benjamini-Hochberg procedure” “false discovery rate (FDR) correction” I suggest calling the procedure in a single name throughout the paper

“Importantly, FDR correction was not applied to pathway and enzyme enrichment analyses since these analyses involve testing the significance of multiple related hypotheses rather than independent hypotheses, which is too conservative, resulting in false negative results.” however the main text mentions p_{adj} . This should be clearly stated in the main text

“performing Spearman correlations of select microbial features” how were the features selected?

Reviewer #3 (Remarks to the Author):

In the manuscript entitled "Gut microbiome remodeling and metabolomic profile in response to multimodal intermittent fasting and higher protein versus calorie restriction", the authors have reported on distinctions observed between an IF-P versus heart-healthy approach to weight loss using diets matched for overall energy intake.

For the study, the authors have used suitable statistical approaches, but interpretation of findings is confounded by use of standardized, isolated/purified fiber supplementation as part of the supplements and snacks consumed on the IF-P regimen. From the standpoint of microbial substrate availability, the IF-P regimen is substantially different from baseline compared to the calorie restricted diet. It is not surprising, therefore, that this produced distinct microbial changes. Because of this, care needs to be taken throughout the manuscript to avoid over-simplification of diet similarity based on broad measures of calories, macronutrients, and grams fiber intake. It is as much a fiber supplementation trial as it is a meal timing trial. The most appropriate approach to this would be a food-based vs nutrient based approach (see Johnson et al. PMID: 31194939).

That said, the effects of the IF-P intervention obviously do not change and are worth the effort the authors put forth in their analysis. However, it is important that the authors make clear throughout that this is a commercial meal replacement shake-based diet versus standard calorie restricted diet. From the standpoint of colonic substrate availability it is misleading to suggest the difference between these diets is simply related to the meal timing. In lieu of pursuing a food-based approach, the authors need to make much more clear throughout that this is not merely a comparison of two whole-food based diets. The observation that certain individuals responded particularly well to the IF-P diet would suggest that maybe insight into responsiveness to the isolated fiber types included in the commercial shakes and snacks may inform future microbiome-based diet/fiber supplementation for weight loss.

Reviewer #4 (Remarks to the Author):

This is an original manuscript by Dr Mohr et al where the authors explore how diets particularly IF-P diet can be leveraged to manipulate the gut microbiome therapeutically for improved gut health and metabolic outcomes with respect to weight loss through a multi-omics approach. The study compared the IF-P diet to CR diet in a randomized control trial over 8 weeks and noted more pronounced effects associated with IF-P diet, with the detection of several unique microbial signatures associated with improved immune responses, metabolomic shifts and weight loss. The authors further followed a participant who showed significant weight loss in response to the IF-P diet after the 8-week intervention using a multi-omics approach and reported stability of the gut microbiome and weight/body composition maintenance up to 1 year. The authors conclude that gut microbiome and metabolome may be involved in maintaining weight loss and body composition.

Although the results on the comparison between the effects of the two different diets on anthropometric and cardiometabolic outcomes have already been published, the originality of this manuscript lies in furthering these studies to explore the interactions between the microbiome,

metabolome and cytokine changes associated with both dietary interventions and the longitudinal tracking of a high responder on an IF-P diet.

The paper holds clinical relevance and is of conceptual interest to several in the field, because it demonstrates how variations in gut microbiome composition and metabolism can shape health outcomes in response to dietary interventions, underscoring the need for microbiome-targeted personalized nutrition in shaping favorable health outcomes. Overall, the manuscript is clear, well-written and concise with data generated from sound methodology. I have few comments for consideration by the authors.

Major comments

The study reports greater microbiome and metabolome shifts in IF-P compared with CR. All participants began with a CR diet for 1 week (run-in period) before the intervention started. This may have biased the findings in favor of IF-P diet since the IF-P intervention diet was markedly different from diet introduced during the run-in period. This point is further supported indirectly from other results where authors do not observe differences between week 4 and 8 in the IF-P group for most assays but differences are mostly noted when weeks 4 and 8 are separately compared with baseline. If the run-in diet was a non-CR diet, would authors expect the same findings or would these outcomes change?

A major claim to some of the observed metabolic effects of IF-P diet is that IF-P diet increased fiber intake. It is unclear to me how that conclusion was reached as I struggle to find that evidence throughout the paper. Can authors elaborate?

Minor comments

Introduction

Line 75 – “the nutritional composition and meal frequency during these periods alters”. “alters” should be “alter”

Line 83-84 – “Thus, the need to examine this in humans is warranted.” “in” should be “is”

Results

Line 116 – “Regarding GI functioning and GM 116 modulation, IF-P significantly decreased sugar and increased dietary fiber relative to CR ($p < 0.05$)”. This section was slightly difficult to understand. I do not see evidence of the results in a table/figure. The supplementary table S1 shows 20-30g of fiber a day for both groups, so it is slightly unclear how to interpret this finding. Can authors provide more information on this was determined since this is a very important finding they highlight throughout the manuscript? Also, I suggest authors report the fiber content of the IF-P in the methods section, just like they have reported for the CR in the methods.

Line 120 – delete “%” after “fat-free mass”

Line 124 – 125: “This finding is likely due to IF-P increasing fiber intake significantly more than CR which may have favorably mediated the GM and symptomatology.” Authors should elaborate on how they arrived at that conclusion.

Line 124 – “... increasing fiber intake "significantly" more.." word should be “significantly”

Line 136-137 – “Figure 1 (g)The GM community structure of IF-P participants shifted significantly throughout the IF-P intervention compared to CR as measured by the Bray-Curtis dissimilarity index.” Were there changes in composition from baseline to weeks 4 and 8 separately for each diet group?

Figure 1h - Is the heatmap showing between group changes for IF-P 0 vs CR 8, for family and genus levels? If correct, what is the basis for that comparison? Why not IF-P at week 4 vs CR at week 4, IF-P week 8 vs CR week 8?

Table 1 legend: “There were no significant differences between groups for GI scores as assessed...” is that correct? The results in the text seem to point to no significance with stool characteristics rather.

Line 186 – “When comparing monozygotic twin pairs, eubacterium ventriosum...” Genus should start with “E”

Line 192- 194: Authors only focus on fecal SCFAs from IF-P. Authors should mention that fecal SCFAs were comparable between both groups

Both diets had 20-30g/day for fiber. Was the fiber make-up/composition the same for both diets?

Line 271 – 272 : “When controlling for these relevant covariates, we observed significant effects of IF-P on 15 metabolites. The results show a log fold change or differential proportions (log (IF-P/CR)) and so these effects occur on CR as well but authors attribute significant effects to only IF-P. The text should reflect that.

Line 338 – “Notably, In the....” . “In” should be “in”

Line 517 - “Although differences between High and Low responders were statistically significant...” I assume the authors meant to say NOT significant rather, correct?

Line 525 – 527: “Under rigorous clinical supervision, this individual was guided through and comprehensively tracked over 52 weeks, strictly adhering to an IF-P regimen, including WL (0-16 weeks) and maintenance (16-52 weeks) periods.” The text appears to suggest there was a maintenance regimen. If that is correct, can authors describe how different the maintenance regimen was from the IF-P regimen?

Line 560 - Figure 6h legend: “Node size corresponds to the proportion of metabolites captured in each pathway set, while node color signifies significance.” What level of significance does each color represent (provide legend for significance)?

Discussion

Line 596: Authors should include space between “of and “concentrations”

Methods

Authors should include fiber details for the IF-P diet like how they indicate that for CR in Line 727.

Shannon index was calculated during the subgroup analysis but not indicated in methods

Statistics

Line 976-979 is repeated at Line 979-982.

Line 1020-1021 - "All statistical tests were performed with a significance level of $p < 0.05$ and Benjamini-Hochberg FDR correction of $p_{adj} \geq 0.10$." p_{adj} should be < 0.10 .

Supplemental information

Figure S5 (d) "Alluvial plot displaying the variation in abundance of the most abundant chemical subclasses ($<1\%$) over time." Authors meant greater than 1% ($>1\%$)

Source/raw data for generating some of the figures have not been provided in the supplementary data. For instance, Figure 1b-g, Figure 2a-d, Fig. S1b-c, Fig. S4a

Other minor comments

Can authors provide information on how compliance to diet was assessed?

"Multimodal" is used in the title but throughout the manuscript a mix of "multi-modal" vs "multimodal" are used. Authors should kindly be consistent.

"Symptomatology" vs "symptomology" – authors should adopt one and be consistent.

Some microbial families are not italicized.

Code availability statement is missing.

STORMS checklist is missing.

REVIEWER COMMENTS

Reviewer #2 (Remarks to the Author):

Review by Saar Shoer of:

"Gut microbiome remodeling and metabolomic profile in response to multimodal intermittent fasting and higher protein versus caloric restriction by Mohr et al.

The study describes microbial and metabolomic signatures between combined intermediate fasting and protein pacing diet versus a calorie-restricted diet, in 20 obese humans over eight weeks. The paper is well written and the figures convey the main message clearly. The study includes various data modalities which is great! However, the reasoning behind using different computational methods to analyze each modality is unclear. The results should be portrayed by the magnitude of change in each diet group rather than focusing on the ones that align with the hypothesis, especially in regard to metabolomic data.

RESPONSE: We are grateful for the constructive feedback and strong support of our manuscript from the Reviewer. The primary rationale for varying computational methods for certain comparisons between diets was to further explore potential mechanisms and the inherent differences in the data types. We have provided additional explanations to improve clarity. We agree with providing a balanced approach for portraying differences for each group regarding metabolomic data presentation. These changes are related below (specific edits and line numbers) and within the accompanying updated manuscript resubmission.

Comments

The title is unclear, especially in "multimodal intermittent fasting and higher protein versus caloric restriction". One can interpret it as three diets. Multimodal in what sense? For example, the description in the abstract is more coherent "between combined IF-P versus ... CR diet" or the one in figure 1 "between intermittent fasting with protein pacing (IF-P) and continuous caloric restriction (CR) diet"

RESPONSE: We have revised the title as suggested (lines 1-2).

Abstract, please also address the results that were in favor of the CR diet, such as the metabolites.

RESPONSE: We've added favorable differences in metabolites for each group to Abstract lines 53-55. In addition, we explain these differences between groups in detail in the main body of the Results section ("IF-P modulates circulating cytokines and gut microbiome taxa compared to CR" and "IF-P and CR yield distinct circulating metabolite signatures and convergence of multiple metabolic pathways") of the manuscript.

Abstract, as n=1 is not a sufficient sample size to draw scientific conclusions, I suggest removing it from the abstract.

RESPONSE: While data from this case study participant is novel and relevant to the overall findings within our manuscript, we agree with the reviewer and have deleted it from the Abstract.

Introduction, "consumed evenly spaced every 3-1/2-4 hours throughout the day", it is unclear what these numbers refer to.

RESPONSE: This sentence has been revised as suggested, lines 79-80.

"We hypothesized an IF-P diet may favorably influence the GM and metabolome" the metabolome results are not in favor of IF-P.

RESPONSE: Based on the findings from our study, the metabolome responses in IF-P were more favorable overall (body composition, microbiome, cardiometabolic health) than CR as related to the primary outcomes of this study. Specifically, we report 15 metabolites were either increased (including, malonic acid, acetylcarnitine; lines 301-313) or decreased (sugar alcohols) which favored improved body composition, cardiometabolic health, gut microbiome, and hunger management in IF-P compared to CR. However, we thoroughly describe the favorable plasma metabolome responses for CR starting at lines 371.

"While several individuals displayed a regression to the mean response", regression to the mean is a mathematical phenomena how can it be attributed to people?

RESPONSE: This has been deleted.

It would be good to mention more study design details in the abstract and introduction, such as 8 weeks, obese participants, n=20, meal form - shakes vs whole foods.

RESPONSE: We agree and have added this at lines 50-51, 84, 92-93.

To me it seems important to mention how many participants dropped out in the main text, rather just in the methods.

RESPONSE: This has been added to the main text in Results section, lines 110-111.

How was the macronutrient distribution of the IF-P diet chosen? "35% carbohydrate, 30% fat, and 110 35% protein"

RESPONSE: This macronutrient distribution was chosen based on our previous investigations (see references 7-10, 15) that used a similar balanced distribution with much success.

Can you estimate how reliable was the monitoring? It seems challenging for an obese person to have 350 kcal per day. One way could be to check if the adherence is in association with the results.

RESPONSE: Given the significant amount of overall weight and body fat loss, especially with IF-P participants in the current and previous RCT's from our laboratory (see refs 8-10), using a similar 36-hour intermittent fasting protocol, the monitoring appears to be highly reliable. Please keep in mind, they were only consuming this amount of calorie intake for one day per week. In addition, only the women IF-P participants were consuming this lower level of intake (350 kcal/day), the men consumed 550 kcals/day for the one (or two) day of intermittent fasting per week.

How were the ranges determined? For example "five to six days per week" and "36-60 hours".

RESPONSE: As stated above, this was based on our previous studies and participants were randomized to either 5 or 6 days of protein pacing and 36 or 60 hours intermittent fasting for the first 4 weeks. This has been added at lines 115-118.

Were participants blinded to the types of arms in the experiment? Did they know if they got the new proposed treatment or the control dietary regime? How could that affect the adherence and results?

RESPONSE: Participants were not blinded and were randomly assigned (see Methods, lines 748-789) to the two types of dietary intervention arms (IF-P vs. CR). However, they received equal support from investigators and registered dietitian (60-minute weekly meetings) throughout the 8-week intervention which resulted in extremely high adherence/compliance of over 90% (only 1 dropout per group) for both groups.

When mentioning p-values, the type of test should be mentioned. Line 114, for example.

RESPONSE: This has been added at lines 120-121.

"This finding is likely due to IF-P increasing fiber intake significantly more than CR which may have favorably mediated the GM and symptomatology." Can you support this claim? Could it be any other key components (lines 319-322) of the interventions that drove these results?

RESPONSE: We apologize for this omission. Fiber has been added to Table 1. However, we agree that other key components referenced and discussed may have also driven these results (lines 226-240).

Please discuss the small number of participants, and how it may affect the results

RESPONSE: Sample size power analysis was based on body weight/composition to achieve an effect size of $F = 0.21$ with 80% power at $\alpha = 0.05$ based on previous data from our lab [see reference 15]. This analysis determined $n = 38$ total sample size was required to detect a significant mean difference

of 1.4-kg weight loss between diet intervention groups (IF-P vs. CR). Thus, our participant size was acceptable. However, we have acknowledged this limitation and how it could influence the results (lines: 707-712).

Figure 1a is great

RESPONSE: Thank you for providing this support of our Results. It's reassuring to receive this level of positive feedback from a reviewer of such a high-quality journal.

Figure 1b, adjusted in what way? Can you discuss the larger variance in the IF-P diet?

RESPONSE: These daily values were adjusted by taking the total weekly intake divided by seven to account for the fasting periods of the IF-P participants, which, by design, contributed to the larger variance. We have added this description to the legend of Figure 1b.

Figure 1d-f, text within subplots is hard to read (time, inter)

RESPONSE: We have increased the font size of this text.

Figure 1e, please define ASV in the legend

RESPONSE: This has been added.

Figure 1f, phylogenetic diversity in which phylogenetic level? species?

RESPONSE: This has been added.

Figure 1h-i, can you translate the beta coefficients to comprehensible units? For example gene copies / g. No species significantly changed in the CR diet? I assume you used linear models, what were they adjusted for? Why not show the direction of change in the bottom panels as in the upper ones?

RESPONSE: To convert beta coefficients into gene copies/g, we would need information about the specific genomes and gene content of the microbial taxa. This information is not available at the genus level in our analysis as we used 16S amplicon sequencing for this section of the paper. Microbial genomes can vary widely within genera, thus trying to assign a uniform value, such as gene copies/g, to all genera may oversimplify the complex biology of these communities. This can introduce errors and inaccuracies into the interpretation. We contend it is more appropriate to focus on the interpretation of beta coefficients in the context of relative abundance changes between the IF-P and CR groups. We would also like to emphasize that our approach is consistent with accepted practices in the field, and members of our group have recently published research in this journal employing a similar methodology (as evidenced by reference to recent publication: <https://www.nature.com/articles/s41467-023-38778-x>).

Linear models were implemented using MaAsLin2 (v1.12.0), a comprehensive R package for efficiently determining multivariable associations between phenotypes, environments, exposures, covariates, and microbial meta'omic features (see <https://huttenhower.sph.harvard.edu/maaslin/>). Our models included group, time, and their interaction, with age and sex as covariates and the participant as a random factor as described in detail in the 'Statistical analysis' section of the Methods.

In relation to the last comment, the panels at the bottom represent p-values, which do not have a direction like beta coefficients.

Can you hypothesize why some species changed in the 4-weeks but not in the 8-weeks? Are the comparisons 4v0 and 8v0 or 4v0 and 8v4?

RESPONSE: As described at lines 193-194 and within Figure 1h-i, these comparisons were made over baseline (i.e., 4v0 and 8v0). The differences in microbial responses between the 4-week and 8-week time points are indeed intriguing and may reflect the complex dynamics of the gut microbiome in response to dietary alterations. Several factors could contribute to this phenomenon, and we added mention of probable community resilience at lines 199-204.

Table 1 "There were no significant differences between groups for GI scores as assessed by Fisher's exact test ($p \geq 0.357$)" at any of the time points? and for non GI scores? such as wet stool weight, bristol stool scale and stool pH? These are likely to be affected by the eating mode
Some of these details are mentioned in the text or methods but it would be better for the figures/tables to be sufficiently informative on their own

RESPONSE: Correct. We have updated the legend for Table 1 including this mention and non-GSRS scores.

"The substantial reduction in calorie intake of both groups (~40% from baseline) did not significantly decrease the level of transient microbial colonization in the gut" did you expect it to decrease? why? did you check whether it significantly changed? decreased or increased?

RESPONSE: We did expect a decrease and provided additional text and a reference at lines 163-175. Please note our models included time, which is already reported within text and in Figure 1.

Please mention how big of a change each diet constitutes relative to the baseline of its participants, and not just between the diets. Is it the same for both diets? If it is greater in the IF-P than in the CR diet, many of the conclusions should be taken with care, such as "a pronounced increase in Bray-Curtis dissimilarity was observed in the IF-P compared to the CR group" and "We observed differential abundance patterns at the family and genus-level in response to the IF-P but not the CR intervention" as these are seemingly the expected results. Could it also be the mode of food intake, shakes vs whole foods, the reason for the differential results?

RESPONSE: We have added the intra-individual median Bray-Curtis dissimilarities for IF-P and CR groups at weeks 4 and 8 and lines 190-192. In addition, although meal replacement shakes were used in IF-P, they consumed at least equal amounts of whole foods (>50%) throughout the 8-week intervention. Regardless, we agree with the reviewer that differences in the mode of calorie intake between groups may have influenced the results, as described in the manuscript (lines 353-55; 646-48).

Lines 222-229, why different methods were used for the anthropometric and microbial analyses?

RESPONSE: The use of distinct analytical approaches for these two domains was based on the specific objectives and nature of the data collected in our investigation. For anthropometric analyses, we employed Spearman correlations due to the simplicity of the data and number of variables. In contrast, for microbial analyses, we employed graph-guided fused least absolute shrinkage and selection operator (GFLASSO) regression. This choice was made considering the intricate interactions within the gut microbiome and the need to identify potential associations between specific microbial taxa and circulating cytokines during the intervention period. GFLASSO regression is particularly suitable for analyzing high-dimensional data, such as microbial composition (as used in a similar fashion: <https://journals.asm.org/doi/full/10.1128/spectrum.00032-22>), and can reveal complex relationships that may not be readily apparent through traditional correlation analyses.

Explain why you focus on fiber intake throughout the paper. For example Lines 124 and 192.

RESPONSE: Fiber intake is frequently considered as a key dietary variable in our findings due both to its relevance in gut microbiome status and the statistically significant increase observed by IF- P relative to CR; nonetheless, we agree with the reviewer there are likely other factors that play a larger role and these are addressed, as stated above.

“While the direction of influence wasn’t clear in this study (e.g., microbes influencing cytokine concentrations), the effect of inflammatory states is well-known to impact gut permeability.” Causality can not be determined in this study setting. The changes in cytokines and microbiome can be independent of each other, and both simply a direct result of the change in diet. In order to support the claims mentioned, one should conduct a mediation analysis to show there is an indirect effect between the two factors .

RESPONSE: We agree that causality cannot be determined in the current study and have added this to the text, line 284.

Bray-Curtis distance was used for the microbiome, while Canberra distance was used for metabolites, why?

RESPONSE: The choice of Bray-Curtis distance for the microbiome data and Canberra distance for the plasma metabolome data was made to align with the nature of the data and the specific goals of each analysis. We chose to use Bray-Curtis distance for the analysis of microbiome data due to its suitability

for compositional data, which is a key characteristic of microbiome abundance data. Bray-Curtis distance accounts for both the presence and relative abundance of microbial taxa within samples, making it well-suited for comparing microbial community composition. In microbiome studies, it is essential to consider not only which taxa are present but also their relative abundances, as shifts in relative abundance can be biologically meaningful. Therefore, Bray-Curtis distance was selected to capture the dissimilarity in microbial community structure between samples.

Conversely, for the analysis of plasma metabolome data, we opted for the Canberra distance metric. Canberra distance is particularly appropriate for continuous data, such as metabolite concentration measurements. In the plasma metabolome, we are primarily interested in the differences in metabolite concentrations between samples. Canberra distance considers the magnitude of differences between data points, which aligns with our objective of assessing dissimilarity in plasma metabolite profiles based on concentration changes.

"100-fold leave-one-out cross-validation" please explain how you ended up with 100 samples at this stage.

RESPONSE: To clarify, the term "100-fold leave-one-out cross-validation" in our manuscript refers to the statistical validation technique we employed, not the number of samples. In this approach, each single sample from our dataset was left out in turn, and the model was trained on the remaining samples. This process was repeated 100 times to ensure robustness and to minimize bias, providing a comprehensive validation of our model. The number '100' here represents the iterations of cross-validation, not the number of samples.

Lines 270-290, the motivation behind this multi-step analysis is unclear, why do the metabolites require a different approach than previously mentioned modalities (such as microbial families or cytokines)?

RESPONSE: The rationale behind this multi-step analysis is rooted in the inherent complexities and characteristics of metabolomic data, as well as our specific research objectives. Consequently, the analysis of metabolomic data often requires tailored approaches to capture the nuanced relationships and interpret complex patterns. We aimed not only to identify metabolites that exhibited significant changes but also to elucidate their potential discriminatory power in distinguishing between the two dietary groups. This objective necessitated a multi-step analysis to progressively refine our understanding of the metabolomic data. The analysis was designed to accomplish the following:

- **Identification of Significant Metabolites:** We initially employed a general linear model (GLM) while controlling for relevant covariates to identify metabolites significantly influenced by the IF-P intervention.
- **Classification Performance Assessment:** To assess the utility of the significant metabolites as potential discriminators between CR and IF-P groups, we conducted a receiver operating characteristic (ROC) analysis.
- **Enhanced Discriminant Analysis:** Given the complexity of the metabolomic data and the aim to improve classification performance, we constructed a supervised orthogonal projection to latent structures discriminant analysis (OPLS-DA) model. This model incorporated the metabolites with the highest variable importance in projection (VIP) scores.

- **Validation and Accuracy:** The refined OPLS-DA model underwent rigorous validation through leave-one-out cross-validation (LOOCV) and permutation testing to ensure its reliability and accuracy in distinguishing between dietary groups.

We believe that this comprehensive analysis strategy was essential for addressing our specific research questions related to dietary intervention effects on circulating metabolite profiles and enhancing the interpretability and discriminatory power of the metabolomic data.

“To determine the significantly impacted pathways of the dietary interventions, we grouped participant samples according to baseline or intervention period (weeks 4 and 8), with IF-P and CR assessed separately.” Again, why is this analysis conducted in a different way than previous ones?

RESPONSE: The specific analysis of impacted pathways was conducted differently for several valid reasons that align with our research objectives. Our study aimed to comprehensively assess the impact of two dietary interventions, intermittent fasting – protein pacing (IF-P) and caloric restriction (CR), on the host's metabolic pathways. These dietary interventions have distinct effects on the host's metabolism (see ref #15), and our objective was to identify and compare the pathways that were significantly impacted by each intervention separately. This separation allowed us to gain insights into the specific metabolic alterations induced by IF-P and CR, enhancing the granularity of our findings. In addition, by analyzing IF-P and CR separately, we were able to identify pathways that exhibited significant alterations in each group, as well as pathways that were common to both groups. This approach provided a nuanced understanding of how these dietary interventions modulate metabolic pathways and whether there are unique or shared metabolic responses. Such insights are valuable for elucidating the mechanisms underlying the observed effects of IF-P and CR on the host's metabolism.

A description of the CR metabolome results is missing

RESPONSE: A detailed description of the CR plasma metabolome results are included beginning at line 361. Indeed, the CR diet appeared to have a pronounced effect on the plasma metabolome overall, whereas the IF-P diet clearly had a more favorable impact on the primary outcomes of the study, as discussed throughout the manuscript and explained above.

Both sections of the results regarding metabolites that are displayed in Figure 3, show greater change in the CR than in the IF-P diet. However, the text is focused on the IF-P group. The authors try to mitigate this point by saying “the greater number of affected pathways and metabolites in the CR group may have been related to the overall greater dietary diversity”. To the reader, it is unclear there is a dietary diversity difference between the groups, and in general this claim does not settle with the microbiome results that were more abundant in the IF-P group

RESPONSE: The reviewer raises a valid point. However, overall, the metabolite differences favored a greater reliance on fat oxidation and less sugar alcohol metabolism in IF-P compared to CR, which align with the significant improvements in GM and greater weight loss and body composition improvements in IF-P study participants. These were the primary outcomes of interest. However, it's

worth noting that CR had favorable changes in certain amino acid metabolites related to longevity and reduction in cancer risk that may or may not be related to dietary diversity. We have revised this in the text at lines 374-376.

“the greater number of affected pathways and metabolites in the CR group may have been related to the overall greater dietary diversity” Can you support this claim? Could it be any other key components (lines 319-322) of the interventions that drove these results?

RESPONSE: We agree with the reviewer and have revised this accordingly.

“we performed a multi-omics factor analysis (MOFA) to identify potential patterns of covariation and co-occurrence between the microbiome and circulating metabolites” Again, why is this analysis conducted in a different way than previous ones?

RESPONSE: The rationale behind conducting MOFA as a separate analysis can be attributed to its unique ability to unveil patterns of covariation and co-occurrence between multi-omic datasets, which aligns with our specific research objectives. This is a well-established tool and was used appropriately as outlined here: <https://biofam.github.io/MOFA2/>

Figure 5 c-d, discuss the large variance in the high and then low groups, respectively

RESPONSE: We agree there appears to be large variances between the groups for both observed species and Shannon index, however, no significant differences were found. This supports our contention as stated at lines 461-465, that most of the variation was explained by the individual, not the group.

Authors say “Assessing metabolite changes between groups did not yield significance when comparing logFC values” meaning no change, and “Pathway analysis of High weight loss responders revealed...” statistical test or significance is not mentioned, but then “Differences detected in our subgroup analysis suggest that the GM composition plays a role in WL responsiveness during IF-P interventions. Notable differences in taxa and fecal metabolites suggest differing substrate utilization capabilities and nutrient-acquiring pathways between High and Low responders, despite being on the same dietary regimen. Although differences between High and Low responders were statistically significant, the magnitude of differences varied, suggesting further research is needed to clarify these differences.” unless I am missing something, the conclusions are not supported by the results

RESPONSE: We have clarified that these differences are referring the microbiome data at lines 552-553.

"Examining positive linear coefficients of a PERMANOVA model, constructed to detect variation between community compositions over time" Again, why use a different method than in the previous species analysis?

RESPONSE: The choice of employing PERMANOVA for this specific case study was driven by its suitability for capturing temporal changes in microbiome composition over a prolonged duration in a single individual. The objective here was to assess how the gut microbiome evolves over an extended period, and PERMANOVA provides a robust approach for analyzing such complex temporal dynamics.

Please use scientific statistical language and methods, line 541 "visually", line 547 "notable", line 574 "observed". It is not clear why these results were described without scientific testing, while the results from line 576 onwards, were analyzed by a statistical method

RESPONSE: In this case study, we recognized the unique nature of the data, where traditional statistical tests were not applicable due to the extended and continuous duration of observations. As such, we employed a visual inspection method to identify notable patterns and trends. Moreover, we are currently analyzing all outcomes in this case study for a subsequent publication and will be highlighting the overall patterns and trends that may inform a larger and longer study.

Line 562, until now, and again in line 568, Canberra distance was used for metabolic distance, why here it is Bray-Curtis?

RESPONSE: These are microbial metabolic pathways. Text has been adjusted.

"after removing the dominant amino acid subclass" why in the analysis displayed in Figure 5g you did not do the same?

RESPONSE: PERMANOVA models were conducted for both removing this subclass (Fig. 6g) and keeping it in (Fig. S5c). This was done to show more nuanced shifts that were obscured from the dominance of this subclass as shown in Fig. S5d).

Lines 584-587, as this section includes a sample size of 1 please refrain from general conclusions

RESPONSE: We have revised this section at line 623-624 to encompass the integrated response from the group comparisons, responders, and case study to capture the conclusions more accurately. We thank the reviewer for this helpful suggestion.

"the IF-P group exhibited a more pronounced community shift and greater divergence from baseline", this is true for some measurements but not for others such as the metabolites. Figure 3 b-c Again, results say "Assessing metabolite changes between groups did not yield significance when

comparing logFC values” so how come the conclusion is “Fecal metabolome analysis further highlighted differences between the two subgroups”

RESPONSE: We have revised this to improve clarity. Here we are referring to GM community shifts as assessed by Bray-Curtis dissimilarity (lines 633-634).

“Notably, the High WL responders displayed enrichment of fecal metabolites involved in lipid metabolism. In contrast, Low responders were more prominent in pathways related to the metabolism of amino acids and peptides, including glycine, serine, and threonine, d-glutamine and d-glutamate, as well as tyrosine metabolism and arginine biosynthesis. The latter metabolic signature has been reported in individuals with severe obesity undergoing high-protein, low-calorie diets.” as commented before, as readers we did not see any scientific analysis to support that

RESPONSE: The observations made regarding the differential metabolic profiles of High and Low responders were based on enrichment analyses. As noted in the text at lines 685-687, we are not implying certain pathways were significant in one group over another. We have added the below after this statement for better context, “Though, as these enrichment analyses were performed in an exploratory manner, we acknowledge the need for a more systematic approach to validate these findings.”

The limitation paragraph in the discussion is good, please add other limitations that came up in this review such as the food intake mode (shakes vs solid food) that could affect among other things gut symptomatology, and the magnitude of dietary nutrients change which could affect the magnitude of change seen in each group. Another limitation is that dietary intake and adherence are self reported “These findings highlight the importance of personalized approaches” can you explain how this study relates to personalization? the connection is unclear

RESPONSE: We are grateful for these additional limitations and have included them in the revised manuscript at lines 717-723. Regarding the findings impact on personalized approaches, a greater understanding of how different dietary regimens impact the microbiome may inform precision nutrition strategies moving forward, especially regarding body composition and cardiometabolic health lifestyle improvements.

Out of curiosity, have you tried comparing the 16S species composition results to those obtained from MetaPhlan

RESPONSE: We have not compared our 16S composition results with the MetaPhlan computational method. However, this remains a valid next step analysis of GM datasets.

I do not have the wet-lab expertise to review the experimental methods, but they seem very detailed. The computational methods on the other hand should be more elaborated

“For MOFA, bacterial 16S rRNA ASVs and plasma metabolites were integrated using the

941 MOFA2 package ... Integrating microbial taxa ... cytokine data and fecal metabolomic data, respectively, was conducted with GFLASSO", please explain the motivation for using different methods

RESPONSE: The motivation behind these discrepancies have been addressed in the above responses.

"After assessing normality (Shapiro-Wilk's tests), LME models were used to test the effect of time and the interaction of group and time with each participant included as a random effect on the alpha diversity metrics using the nLME package. For beta diversity, a nested permutational analysis of variance (PERMANOVA) was conducted on Bray-Curtis dissimilarities using the Adonis test in the vegan package (v2.6.2) with permutations." why use different methods?

RESPONSE: This has been addressed in response to your earlier comments.

I would split the computational methods description, as right now, if someone is looking for details of a specific analysis, one need to read the entire section

RESPONSE: We respectfully disagree. Bioinformatic description has been relayed in the appropriate sections, including 'Fecal microbiome analysis', 'Targeted plasma metabolomic analysis', 'Multi-omics data analysis', etc. Statistical assessment of each data set flows logically with the order of the paper and is presented in preferred fashion for this journal.

"To detect changes in microbial features between groups over time, we built linear mixed models" and "Univariate and multivariate analyses of plasma metabolites and metabolic ontology analysis were performed" why use different methods?

RESPONSE: It is well understood different omic data types require different handling, statistically.

"The Benjamini-Hochberg procedure" "false discovery rate (FDR) correction" I suggest calling the procedure in a single name throughout the paper

RESPONSE: We have adjusted this throughout.

"Importantly, FDR correction was not applied to pathway and enzyme enrichment analyses since these analyses involve testing the significance of multiple related hypotheses rather than independent hypotheses, which is too conservative, resulting in false negative results." however the main text mentions p.adj. This should be clearly stated in the main text

RESPONSE: As stated, this was for the fecal metabolome, not the plasma metabolome where we assessed the entire study cohort. Regardless, we added additional text at line 1071 for the reader

stating this was for the subgroup analysis.

“performing Spearman correlations of select microbial features” how were the features selected?

RESPONSE: As mentioned in this sentence, from the “highest beta coefficients detected from GFLASSO models”. We have added the cutoff value at line 1077.

Reviewer #3 (Remarks to the Author):

In the manuscript entitled "Gut microbiome remodeling and metabolomic profile in response to multimodal intermittent fasting and higher protein versus calorie restriction", the authors have reported on distinctions observed between an IF-P versus heart-healthy approach to weight loss using diets matched for overall energy intake.

For the study, the authors have used suitable statistical approaches, but interpretation of findings is confounded by use of standardized, isolated/purified fiber supplementation as part of the supplements and snacks consumed on the IF-P regimen. From the standpoint of microbial substrate availability, the IF-P regimen is substantially different from baseline compared to the calorie restricted diet. It is not surprising, therefore, that this produced distinct microbial changes. Because of this, care needs to be taken throughout the manuscript to avoid over-simplification of diet similarity based on broad measures of calories, macronutrients, and grams fiber intake. It is as much a fiber supplementation trial as it is a meal timing trial. The most appropriate approach to this would be a food-based vs nutrient based approach (see Johnson et al. PMID: 31194939).

That said, the effects of the IF-P intervention obviously do not change and are worth the effort the authors put forth in their analysis. However, it is important that the authors make clear throughout that this is a commercial meal replacement shake-based diet versus standard calorie restricted diet.

RESPONSE: We are grateful for the Reviewers thoughtful critique of our manuscript. Specifically, regarding the fiber supplementation in the meal replacement shakes consumed by IF-P participants, these contained ~9 grams of fiber per serving which they consumed twice a day, for a total of 18 grams. Please note, both groups were consuming 20 and 24 grams of fiber/day at baseline (IF-P and CR respectively), in their normal diets. We have now added this text at lines 123 stating that at the end of the 8-week intervention, IF-P significantly increased fiber intake (26 grams/day) vs. CR (24 grams/day); however, as we noted above, this level of change may not have played a significant role in GM differences between groups, as stated at lines 222-23. It's also important to highlight that IF-P was consuming more than 50% of their total calorie intake as whole foods, of similar nature to what CR participants were consuming. This included, whole grains, fresh fruits and vegetables, healthy fats, and high-quality plant/animal-based proteins.

From the standpoint of colonic substrate availability it is misleading to suggest the difference between these diets is simply related to the meal timing.

RESPONSE: We state at lines 72-74, “... One of the main processes by which the GM affects host physiology is producing bioactive metabolites from the gastrointestinal (GI) contents. Nutrient composition, feeding frequency, and meal timing impact this dependency.” Thus, it’s our contention that meal timing is one of several contributing factors for differences in GM diversity.

In lieu of pursuing a food-based approach, the authors need to make much more clear throughout that this is not merely a comparison of two whole-food based diets.

RESPONSE: Throughout the manuscript, we highlight the inclusion of meal replacement shakes (lines 83, 229, 353-354, 647, 718, 770, 776). Moreover, please note, the IF-P participants consumed at least 50% and as much as 75% of their total daily calorie intake as whole-foods. The meal replacement shakes were used primarily to support protein pacing and do not provide the majority of calories consumed by IF-P participants.

The observation that certain individuals responded particularly well to the IF-P diet would suggest that maybe insight into responsiveness to the isolated fiber types included in the commercial shakes and snacks may inform future microbiome-based diet/fiber supplementation for weight loss.

RESPONSE: We agree with the Reviewer that isolated fiber types within commercial shakes and snacks may be a valuable strategy to modify microbiome-based weight loss. As stated above in the first response to the Reviewer, the total fiber only increased 6 grams from pre to post in the IF-P group and both groups consumed similar amounts of fiber each day (24-26 grams/day) throughout the 8-week intervention. Further, as stated at lines 227-230, the type (resistant starch) and amount of fiber provided by the meal replacement shakes (18 grams) was less than amounts previously shown (>20 grams/day) to favorably modulate short chain fatty acid production (lines 230-231). Thus, it’s unlikely that fiber intake had a major influence on microbiome responsiveness between groups. However, future studies should consider fiber types with microbiome-based weight loss.

Reviewer #4 (Remarks to the Author):

This is an original manuscript by Dr Mohr et al where the authors explore how diets particularly IF-P diet can be leveraged to manipulate the gut microbiome therapeutically for improved gut health and metabolic outcomes with respect to weight loss through a multi-omics approach. The study compared the IF-P diet to CR diet in a randomized control trial over 8 weeks and noted more pronounced effects associated with IF-P diet, with the detection of several unique microbial signatures associated with improved immune responses, metabolomic shifts and weight loss. The authors further followed a participant who showed significant weight loss in response to the IF-P diet after the 8-week intervention using a multi-omics approach and reported stability of the gut microbiome and weight/body composition maintenance up to 1 year. The authors conclude that gut microbiome and metabolome may be involved in maintaining weight loss and body composition.

Although the results on the comparison between the effects of the two different diets on anthropometric and cardiometabolic outcomes have already been published, the originality of this manuscript lies in furthering these studies to explore the interactions between the microbiome, metabolome and cytokine changes associated with both dietary interventions and the longitudinal tracking of a high responder on an IF-P diet.

The paper holds clinical relevance and is of conceptual interest to several in the field, because it demonstrates how variations in gut microbiome composition and metabolism can shape health outcomes in response to dietary interventions, underscoring the need for microbiome-targeted personalized nutrition in shaping favorable health outcomes. Overall, the manuscript is clear, well-written and concise with data generated from sound methodology. I have few comments for consideration by the authors.

RESPONSE: Thank you for the positive support of our manuscript and especially acknowledging the novelty and relevance of our findings to advance the field of microbiome-targeted personalized nutrition to enhance health outcomes, as well as the clarity of the writing and methodology.

Major comments

The study reports greater microbiome and metabolome shifts in IF-P compared with CR. All participants began with a CR diet for 1 week (run-in period) before the intervention started. This may have biased the findings in favor of IF-P diet since the IF-P intervention diet was markedly different from diet introduced during the run-in period. This point is further supported indirectly from other results where authors do not observe differences between week 4 and 8 in the IF-P group for most assays but differences are mostly noted when weeks 4 and 8 are separately compared with baseline. If the run-in diet was a non-CR diet, would authors expect the same findings or would these outcomes change?

RESPONSE: The 1-week run-in period was a baseline control in which participants maintained a stable body weight by consuming a similar caloric intake as their pre-enrollment caloric intake while maintaining their sedentary lifestyle. Thus, the run-in diet was a non-CR (as well as non-IF-P) diet and was therefore, neutral for both groups.

A major claim to some of the observed metabolic effects of IF-P diet is that IF-P diet increased fiber intake. it is unclear to me how that conclusion was reached as I struggle to find that evidence throughout the paper. Can authors elaborate?

RESPONSE: We agree with the Reviewer and have revised the manuscript accordingly and deleted/modified any references to fiber influencing metabolic effects.

Minor comments

Introduction

Line 75 – “the nutritional composition and meal frequency during these periods alters”. "alters should be “alter”

RESPONSE: Revised.

Line 83-84 – “Thus, the need to examine this in humans in warranted.” "in" should be “is”

RESPONSE: Revised.

Results Line 116 – “Regarding GI functioning and GM 116 modulation, IF-P significantly decreased sugar and increased dietary fiber relative to CR ($p < 0.05$)”. This section was slightly difficult to understand. I do not see evidence of the results in a table/figure. The supplementary table S1 shows 20-30g of fiber a day for both groups, so it is slightly unclear how to interpret this finding. Can authors provide more information on this was determined since this is a very important finding they highlight throughout the manuscript? Also, I suggest authors report the fiber content of the IF-P in the methods section, just like they have reported for the CR in the methods.

RESPONSE: We have added the fiber content at line 775 of the Methods.

Line 120 – delete “%” after “fat-free mass”

RESPONSE: Revised.

Line 124 – 125: “This finding is likely due to IF-P increasing fiber intake significantly more than CR which may have favorably mediated the GM and symptomatology.” Authors should elaborate on how they arrived at that conclusion.

RESPONSE: We agree with the Reviewer and have revised the sentence to reflect the differences in protein and sugar intake between the two groups, instead of fiber intake.

Line 124 – “... increasing fiber intake “significantly” more..” word should be “significantly”

RESPONSE: Revised.

Line 136-137 – “Figure 1 (g)The GM community structure of IF-P participants shifted significantly throughout the IF-P intervention compared to CR as measured by the Bray-Curtis dissimilarity index.” Were there changes in composition from baseline to weeks 4 and 8 separately for each diet group?

RESPONSE: Yes, as stated in the text and Figure 1g legend, the changes in Bray-Curtis dissimilarity index were greater for IF-P than CR, as measured by Wilcoxon rank-sum test.

Figure 1h - Is the heatmap showing between group changes for IF-P 0 vs CR 8, for family and genus levels? If correct, what is the basis for that comparison? Why not IF-P at week 4 vs CR at week 4, IF-P week 8 vs CR week 8?

RESPONSE: Figure 1h are heatmap comparisons for family and Figure 1i are for genus, as stated in the Figure 1 legend at lines 149-152. Indeed, the heatmaps show the comparison between IF-P vs. CR at week 0-4, upper panel, and IF-P vs. CR at week 0-8, lower panel.

Table 1 legend: "There were no significant differences between groups for GI scores as assessed..." is that correct? The results in the text seem to point to no significance with stool characteristics rather.

RESPONSE: No differences existed between groups for GI and stool characteristics, as stated at lines 172-175. However, IF-P improved GI symptom ratings at each time point compared to CR.

Line 186 – "When comparing monozygotic twin pairs, eubacterium ventriosum..." Genus should start with "E"

RESPONSE: Revised.

Line 192- 194: Authors only focus on fecal SCFAs from IF-P. Authors should mention that fecal SCFAs were comparable between both groups

RESPONSE: We agree and have revised this sentence at lines 232-233.

Both diets had 20-30g/day for fiber. Was the fiber make-up/composition the same for both diets?

RESPONSE: Differences in fiber make-up/composition between groups was not a focus of the study and total fiber was similar between groups throughout the 8-week intervention. IF-P participants consumed up to two meal replacement shakes per day, each containing ~9 grams of fiber (line 768), the remaining fiber (~10 grams) was consumed from whole foods. CR participants consumed their fiber via whole food sources.

Line 271 – 272 : "When controlling for these relevant covariates, we observed significant effects of IF-P on 15 metabolites. The results show a log fold change or differential proportions (log (IF-P/CR)) and so these effects occur on CR as well but authors attribute significant effects to only IF-P. The text should reflect that.

RESPONSE: We agree and have adjusted this sentence (lines 300-302) to now read as, "When controlling for these relevant covariates, we observed significant differences between IF-P and CR for 15 metabolites."

Line 338 – "Notably, In the..." . "In" should be "in"

RESPONSE: Revised.

Line 517 - "Although differences between High and Low responders were statistically significant..." I assume the authors meant to say NOT significant rather, correct?

RESPONSE: This sentence has been revised to reflect the differences in the microbiome data.

Line 525 – 527: "Under rigorous clinical supervision, this individual was guided through and comprehensively tracked over 52 weeks, strictly adhering to an IF-P regimen, including WL (0-16 weeks) and maintenance (16-52 weeks) periods." The text appears to suggest there was a maintenance regimen.

If that is correct, can authors describe how different the maintenance regimen was from the IF-P regimen?

RESPONSE: Weight maintenance was established by maintaining energy balance. This has been revised at line 563.

Line 560 - Figure 6h legend: "Node size corresponds to the proportion of metabolites captured in each pathway set, while node color signifies significance." What level of significance does each color represent (provide legend for significance)?

RESPONSE: We have added a legend to the figure panel.

Discussion

Line 596: Authors should include space between "of and "concentrations"

RESPONSE: Revised.

Methods

Authors should include fiber details for the IF-P diet like how they indicate that for CR in Line 727.

RESPONSE: This has been added at lines 771 and 775.

Shannon index was calculated during the subgroup analysis but not indicated in methods

RESPONSE: This has been added to the 'Fecal microbiome analysis' in the Methods section.

Statistics

Line 976-979 is repeated at Line 979-982.

RESPONSE: The repeat has been removed.

Line 1020-1021 - "All statistical tests were performed with a significance level of $p < 0.05$ and Benjamini-Hochberg FDR correction of $p_{adj} \geq 0.10$." p_{adj} should be < 0.10 .

RESPONSE: Revised.

Supplemental information

Figure S5 (d) "Alluvial plot displaying the variation in abundance of the most abundant chemical subclasses (<1%) over time." Authors meant greater than 1% (>1%)

RESPONSE: Revised.

Source/raw data for generating some of the figures have not been provided in the supplementary data. For instance, Figure 1b-g, Figure 2a-d, Fig. S1b-c, Fig. S4a

RESPONSE: This data has now been included.

Other minor comments

Can authors provide information on how compliance to diet was assessed?

RESPONSE: This information is included at lines 762-766 Briefly, compliance was monitored by daily food record monitoring by the research team and weekly registered dietitian meetings.

"Multimodal" is used in the title but throughout the manuscript a mix of "multi-modal" vs "multimodal" are used. Authors should kindly be consistent.

RESPONSE: We apologize for the inconsistency and have revised it to "multimodal" throughout.

"Symptomatology" vs "symptomology" – authors should adopt one and be consistent.

RESPONSE: Revised to symptomatology throughout.

Some microbial families are not italicized.

RESPONSE: This has been corrected throughout all submitted documents.

Code availability statement is missing.

RESPONSE: This has been added and scripts will be added prior to publication.

STORMS checklist is missing.

RESPONSE: This has been added, with some minor adjusts throughout the manuscript to fully adhere to the guidelines of STORMS. In the checklist document, please note that page and line numbers correspond to the marked-up version of this resubmission.

REVIEWER COMMENTS

Reviewer #2 (Remarks to the Author):

Principle study limitations that are still not acknowledged:

Magnitude of change relative to the baseline diet is different for each arm, thus difference in magnitude of effect is expected. I raised this point multiple times, however it is still not addressed. Reviewer #3 agrees “the IF-P regimen is substantially different from baseline compared to the calorie restricted diet”. Reviewer #4 also agrees “This may have biased the findings in favor of IF-P diet since the IF-P intervention diet was markedly different from diet introduced during the run-in period.”

I suggest simply showing a scatterplot for the main dietary components (carbohydrates, lipids, proteins and fibers), where each participant is a dot colored by the intervention arm, the x-axis the run-in dietary value, the y-axis the intervention dietary value.

Participants were not blinded to dietary arm assignment, which may affect their conscious and unconscious behavior and thus response to measured outcomes.

The results should be portrayed by the magnitude of change in each diet group rather than focusing on the ones that align with the hypothesis, especially in regard to metabolomic data.

Cytokines CR diet results are not reported, unless I am missing something. Abstract and discussion makes it sound like they were tested in both groups and results were only found in the IF-P diet.

Can you estimate how reliable was the monitoring? It seems challenging for an obese person to have 350 kcal per day. One way could be to check if the adherence is in association with the results.

RESPONSE: Given the significant amount of overall weight and body fat loss, especially with IF-P participants in the current and previous RCT's from our laboratory (see refs 8-10), using a similar 36-hour intermittent fasting protocol, the monitoring appears to be highly reliable. Please keep in mind, they were only consuming this amount of calorie intake for one day per week. In addition, only the women IF-P participants were consuming this lower level of intake (350 kcal/day), the men consumed 550 kcal/day for the one (or two) day of intermittent fasting per week.

Participants were not blinded to arm assignment, which may have caused different levels of adherence in each group. I suggest you show this is not the case or acknowledge it is.

Were participants blinded to the types of arms in the experiment? Did they know if they got the new proposed treatment or the control dietary regime? How could that affect the adherence and results?

RESPONSE: Participants were not blinded and were randomly assigned (see Methods, lines 748-789) to the two types of dietary intervention arms (IF-P vs. CR). However, they received equal support from investigators and registered dietitian (60-minute weekly meetings) throughout the 8-week

intervention which resulted in extremely high adherence/compliance of over 90% (only 1 dropout per group) for both groups.

Please do not confuse adherence with completion. Among the participants who completed the study there sure are different levels of adherence to the guidelines.

“This finding is likely due to IF-P increasing fiber intake significantly more than CR which may have favorably mediated the GM and symptomatology.” Can you support this claim? Could it be any other key components (lines 319-322) of the interventions that drove these results?

RESPONSE: We apologize for this omission. Fiber has been added to Table 1. However, we agree that other key components referenced and discussed may have also driven these results (lines 226-240).

What about the liquid parts of the IF-P diet?

In relation to the last comment, the panels at the bottom represent p-values, which do not have a direction like beta coefficients.

Why are the coefficients in the bottom panel of Figure 1h not shown as in the upper one?

Can you hypothesize why some species changed in the 4-weeks but not in the 8-weeks? Are the comparisons 4v0 and 8v0 or 4v0 and 8v4?

RESPONSE: As described at lines 193-194 and within Figure 1h-i, these comparisons were made over baseline (i.e., 4v0 and 8v0). The differences in microbial responses between the 4-week and 8-week time points are indeed intriguing and may reflect the complex dynamics of the gut microbiome in response to dietary alterations. Several factors could contribute to this phenomenon, and we added mention of probable community resilience at lines 199-204.

Community resilience is an interesting suggestion.

Reviewer #4 has suggested a possible reason for seeing results in the IF-P diet and not in the CR diet - if the run-in period is more similar to the CR diet than it is to the IF-P diet, it introduces a major bias to the results, and the comparison between them should at least in the text address that. In line with my comments on the subject.

Please mention how big of a change each diet constitutes relative to the baseline of its participants, and not just between the diets. Is it the same for both diets? If it is greater in the IF-P than in the CR diet, many of the conclusions should be taken with care, such as “a pronounced increase in Bray-Curtis dissimilarity was observed in the IF-P compared to the CR group” and “We observed differential abundance patterns at the family and genus-level in response to the IF-P but not the CR intervention” as these are seemingly the expected results. Could it also be the mode of food intake, shakes vs whole foods, the reason for the differential results?

RESPONSE: We have added the intra-individual median Bray-Curtis dissimilarities for IF-P and CR groups at weeks 4 and 8 and lines 190-192. In addition, although meal replacement shakes were used in IF-P, they consumed at least equal amounts of whole foods (>50%) throughout the 8-week intervention. Regardless, we agree with the reviewer that differences in the mode of calorie intake

between groups may have influenced the results, as described in the manuscript (lines 353-55; 646-48).

I ask about diet, the response is about microbiome. This is a main point regarding the study design the authors do not address.

“While the direction of influence wasn’t clear in this study (e.g., microbes influencing cytokine concentrations), the effect of inflammatory states is well-known to impact gut permeability.” Causality can not be determined in this study setting. The changes in cytokines and microbiome can be independent of each other, and both simply a direct result of the change in diet. In order to support the claims mentioned, one should conduct a mediation analysis to show there is an indirect effect between the two factors .

RESPONSE: We agree that causality cannot be determined in the current study and have added this to the text, line 284.

If not conducting a mediation analysis, please acknowledge the microbiome and cytokines correlated effect can both be a direct result of the diet, in the results section and the discussion part pointing to these results.

“100-fold leave-one-out cross-validation” please explain how you ended up with 100 samples at this stage.

RESPONSE: To clarify, the term "100-fold leave-one-out cross-validation" in our manuscript refers to the statistical validation technique we employed, not the number of samples. In this approach, each single sample from our dataset was left out in turn, and the model was trained on the remaining samples. This process was repeated 100 times to ensure robustness and to minimize bias, providing a comprehensive validation of our model. The number '100' here represents the iterations of cross-validation, not the number of samples.

K-fold is a term used for splitting the data to K equal size parts. Leave-one-out is when you take one sample out for a test set, and do it for all the samples. Please rephrase to make it clear it is 100 repetitions and not splitting the data to 100 equal size parts.

Both sections of the results regarding metabolites that are displayed in Figure 3, show greater change in the CR than in the IF-P diet. However, the text is focused on the IF-P group. The authors try to mitigate this point by saying “the greater number of affected pathways and metabolites in the CR group may have been related to the overall greater dietary diversity”. To the reader, it is unclear there is a dietary diversity difference between the groups, and in general this claim does not settle with the microbiome results that were more abundant in the IF-P group

RESPONSE: The reviewer raises a valid point. However, overall, the metabolite differences favored a greater reliance on fat oxidation and less sugar alcohol metabolism in IF-P compared to CR, which align with the significant improvements in GM and greater weight loss and body composition improvements in IF-P study participants. These were the primary outcomes of interest. However, it’s worth noting that CR had favorable changes in certain amino acid metabolites related to longevity and reduction in cancer risk that may or may not be related to dietary diversity. We have revised this in the text at lines 374-376.

To the reader, it is still unclear there is a dietary diversity difference between the groups.

Please use scientific statistical language and methods, line 541 “visually”, line 547 “notable”, line 574 “observed”. It is not clear why these results were described without scientific testing, while the results from line 576 onwards, were analyzed by a statistical method

RESPONSE: In this case study, we recognized the unique nature of the data, where traditional statistical tests were not applicable due to the extended and continuous duration of observations. As such, we employed a visual inspection method to identify notable patterns and trends. Moreover, we are currently analyzing all outcomes in this case study for a subsequent publication and will be highlighting the overall patterns and trends that may inform a larger and longer study.

“we provide evidence of long-term GM stabilization from these changes by following one individual over 12 months” as the case study is not statistically tested I would not call it “evidence”.

“Pathway analysis identified significant impacts” the p-values are not corrected for multiple hypothesis testing so I would not call it “significant”.

The limitation paragraph in the discussion is good, please add other limitations that came up in this review such as the food intake mode (shakes vs solid food) that could affect among other things gut symptomatology, and the magnitude of dietary nutrients change which could affect the magnitude of change seen in each group. Another limitation is that dietary intake and adherence are self reported

“These findings highlight the importance of personalized approaches” can you explain how this study relates to personalization? the connection is unclear

RESPONSE: We are grateful for these additional limitations and have included them in the revised manuscript at lines 717-723. Regarding the findings impact on personalized approaches, a greater understanding of how different dietary regimens impact the microbiome may inform precision nutrition strategies moving forward, especially regarding body composition and cardiometabolic health lifestyle improvements.

The magnitude of dietary nutrients change which could affect the magnitude of change seen in each group is still not addressed.

Reviewer #3 (Remarks to the Author):

Previous concerns regarding interpretation of dietary fiber intake persist. The authors continue to use dietary fiber too comprehensively given the nature of the IF-P intervention. Fiber from whole foods exists in a complex matrix whereas fiber from isolated sources produces distinct effects on the microbiome. In the case of the IF-P arm, while grams of dietary fiber were, for all intents and purposes, similar to baseline and the CR arm, the introduction of 18g (70% of total IF-P arm fiber

intake) being from an isolated fiber source would be expected to produce distinct effects (See PMID: 32846882 for limitations of the generic term "dietary fiber"). Given the nature of the intervention arms and study design, the IF-P arm vs CR arm was similar to an isolated fiber supplement study (e.g. PMID: 37444219). This needs to be given more consideration and appreciation when comparing the study arms and interpreting results.

Notably, however, the effects of the IF-P intervention are still impressive and worth the continued effort the authors have put forth in their analysis.

Reviewer #4 (Remarks to the Author):

Thank you for the opportunity to review a revised manuscript entitled "Gut microbiome remodeling and metabolomic profile in response to protein pacing with intermittent fasting versus continuous caloric restriction" by Dr Mohr et al. The authors have made significant efforts to address reviewer comments.

There are few minor comments:

Figure 1 shows most analysis where data in weeks 4 and 8 are compared to baseline (week 0) data except for Fig.1g (community composition) where the IF-P at weeks 4 and 8 is compared to CR and not a shift from baseline to the other weeks separately for IF-P and CR. Basically week 0 is missing. Can authors include time point 0 to Fig.1g to conform with the rest of the sub figures in Fig.1 or is there a reason why that was not included? Especially when authors refer to a shift from baseline to other time points in their results section in line 185-186 ("To understand the taxa driving this GM variation from baseline to weeks four and eight between the two dietary interventions...") yet the data/figure is not represented.

Fig5, ie line 507 (5j) legend reads: "Grid-fused least absolute shrinkage and selection operator (LASSO) regression" - is it GFLASSO or LASSO? Also check the legend for Figures2 in the supplemental information, it also has LASSO.

Line 664-665 in the discussion reads: "In contrast, the Low-responder group exhibited a decreased abundance of butyrate-producing bacteria."

The results in Fig. 5e does not appear to suggest that, if anything at all, there is comparable enrichment in the abundance of butyrate producing bacteria for both low and high responders. Can authors explain how they arrive at that conclusion? since I find that text being somehow overstated.

It appears the line numbers in the STORMS checklist do not match with the main document (pdf version). Here are 3 examples under Methods section:

Study Design: Authors state line 95, I find that information in line 92.

Antibiotics usage: Authors state line 743, that information appears in a different line number.

Ethics: Authors state line 749-752, information is not available in the stated line numbers.

Can authors recheck the whole checklist to make sure all items conform to line numbers in the main text?

Line 116 ANOVA's should be "ANOVA"

Delete the empty table after TableS1 in supplemental information

Reviewer #4 (Remarks on code availability):

The account has been created but the code is not available for review.

REVIEWER COMMENTS

Reviewer #2 (Remarks to the Author):

Principle study limitations that are still not acknowledged:

Magnitude of change relative to the baseline diet is different for each arm, thus difference in magnitude of effect is expected. I raised this point multiple times, however it is still not addressed. Reviewer #3 agrees “the IF-P regimen is substantially different from baseline compared to the calorie restricted diet”. Reviewer #4 also agrees “This may have biased the findings in favor of IF-P diet since the IF-P intervention diet was markedly different from diet introduced during the run-in period.”

I suggest simply showing a scatterplot for the main dietary components (carbohydrates, lipids, proteins and fibers), where each participant is a dot colored by the intervention arm, the x-axis the run-in dietary value, the y-axis the intervention dietary value.

We are grateful for Reviewer 2’s comprehensive and insightful comments of our manuscript.

With specific regard to the reviewer’s primary concern of the “magnitude of change relative to baseline diet is different for each arm, thus difference in magnitude of effect is expected”. Please note, by methodological study design, a) the baseline run-in diets were identical between groups, b) the intervention arm diets for both IF-P and CR were significantly different from baseline run-in diets, and c) the intervention arm diets were significantly different from each other. These basic study design features are hallmarks of well-controlled randomized clinical trials using a nutritional intervention. We’ve provided detailed figures (Fig 1b, Supplemental Fig S1b), tables (Supplemental Data – 1), and reference #15 in the original submission. Reviewers 3 and 4 have not expressed these concerns in their latest comments. However, based on previous feedback from the other reviewers on this issue, we’ve highlighted no baseline differences between groups for any dietary intake variable at lines 114-116, as well as shown in Figure S1b. Please note, despite similar baseline dietary intake for all variables, intervention groups (IF-P and CR) were significantly different from baseline and between each other during the intervention, as designed.

Participants were not blinded to dietary arm assignment, which may affect their conscious and unconscious behavior and thus response to measured outcomes.

Correct. This is an assumed and accepted limitation with human RCT nutrition research. Please see: PMID: 28893297; PMCID: PMC5594518.

The results should be portrayed by the magnitude of change in each diet group rather than focusing on the ones that align with the hypothesis, especially in regard to metabolomic data.

Cytokines CR diet results are not reported, unless I am missing something. Abstract and discussion makes it sound like they were tested in both groups and results were only found in the IF-P diet.

In our manuscript, we aimed to highlight the significant interaction effects and subsequent pairwise comparisons that revealed changes in cytokine levels. Indeed, both the IF-P and CR groups were tested for changes in cytokine expression. However, upon re-examination of our presentation of these results, we recognize that the narrative may have inadvertently emphasized the findings from the IF-P group, potentially leading to ambiguity regarding the analysis and results for the CR group. We have now added a summary of the cytokine levels in the CR group, explicitly stating that no significant changes were observed in the tested cytokines (IL-4, IL-6, IL-8, IL-13) over the study period (see lines 273-274). As before, these results are displayed in Fig. 2.

Can you estimate how reliable was the monitoring? It seems challenging for an obese person to have 350 kcal per day. One way could be to check if the adherence is in association with the results.

RESPONSE: Given the significant amount of overall weight and body fat loss, especially with IF-P participants in the current and previous RCT's from our laboratory (see refs 8-10), using a similar 36-hour intermittent fasting protocol, the monitoring appears to be highly reliable. Please keep in mind, they were only consuming this amount of calorie intake for one day per week. In addition, only the women IF-P participants were consuming this lower level of intake (350 kcal/day), the men consumed 550 kcals/day for the one (or two) day of intermittent fasting per week.

Participants were not blinded to arm assignment, which may have caused different levels of adherence in each group. I suggest you show this is not the case or acknowledge it is.

Correct, the participants were not blinded, and although the compliance was >90%, we are not able to assume adherence was the same. However, in the CONSORT, we include compliance rates, and it's assumed under positive behavior change that compliance and adherence are similar. As previously stated, we only had 1 participant per group not complete the study. Regarding adherence, this was reflected by each study participant adhering to more than 90% for each of the following study requirements: a) completion of the 2-day food diary analyses, b) weekly dietary intake journal inspections, c) attending weekly meetings with the dietitian, d) attending meal distributions and returning empty containers (see ref. 15, original study publication). The levels at which these were completed pertain to both compliance and adherence of participants in each group, as opposed to how many finished the trials. Adherence and compliance would be encouraged through clear written and verbal instructions, daily contact with team members as well as weekly contact with the dietitian, all of which occurred in the current study. According to this review:

<https://www.ncbi.nlm.nih.gov/pmc/articles/PMC10324868/>, adherence is the extent to which a person's behavior, following a diet, or executing other lifestyle changes correspond with agreed upon recommendations from a health care provider, or researcher in this case, it's an active choice.

Compliance on the other hand measures the extent to which their behavior matches the recommendations or following instructions (passive).

Were participants blinded to the types of arms in the experiment? Did they know if they got the new proposed treatment or the control dietary regime? How could that affect the adherence and results?

RESPONSE: Participants were not blinded and were randomly assigned (see Methods, lines 748-789) to the two types of dietary intervention arms (IF-P vs. CR). However, they received equal support from investigators and registered dietitian (60-minute weekly meetings) throughout the 8-week intervention

which resulted in extremely high adherence/compliance of over 90% (only 1 dropout per group) for both groups.

Please do not confuse adherence with completion. Among the participants who completed the study there sure are different levels of adherence to the guidelines.

We agree there are subtle differences between adherence and compliance. While participants adhered to the study requirements (>90%), as stated above, there is the assumption they were compliant as well.

“This finding is likely due to IF-P increasing fiber intake significantly more than CR which may have favorably mediated the GM and symptomatology.” Can you support this claim? Could it be any other key components (lines 319-322) of the interventions that drove these results?

RESPONSE: We apologize for this omission. Fiber has been added to Table 1. However, we agree that other key components referenced and discussed may have also driven these results (lines 226-240).

What about the liquid parts of the IF-P diet?

We appreciate the reviewer bringing this to our attention. We have elaborated on the liquid meal replacements in the IF-P diet, highlighting how consumption may lead to the rapid and direct delivery of fiber and other nutrients to the gastrointestinal tract, in contrast to CR participants, who introduce a complex fiber profile more slowly through the consumption of solid meals (lines 230-242). We note that the variance in the rate of nutrient delivery may be an influencer of gut microbial composition and structure, notwithstanding the similarities we noted in SCFA biosynthesis. Furthermore, we acknowledge that the investigation into the effects of liquid versus solid dietary components on gut health is an emerging area of research. Additionally, the popularity and effectiveness of liquid-based meal replacement products is supported by the published literature suggesting greater adherence due to simple meals in pre-defined serving sizes. They also may aid weight loss to a greater degree than caloric restriction as well as weight maintenance. Thus, the current study was well-designed and in a way that aligns with consumer dietary preferences and practices (see also; PMID: 12704397, PMID: 11269616, PMID: 23236298). Therefore, we underscore the necessity for future studies to delve into the differential impacts of these dietary forms on the gut microbiome and SCFA production.

In relation to the last comment, the panels at the bottom represent p-values, which do not have a direction like beta coefficients.

Why are the coefficients in the bottom panel of Figure 1h not shown as in the upper one?

The design of Figure 1h was considered to best communicate the complex dataset involving changes in gut microbiome taxa over time across different groups. The upper panel of the figure uses color gradients to visually convey the magnitude of change within each group, with asterisks indicating statistical significance. The bottom panel is designed to emphasize the significance of between-group differences in changes over time (which can be inferred from the above panel). The Black-White annotations serve this purpose effectively, directing the reader's attention to the comparative aspect of the data, which is the panel's primary focus. Our design choices align with best practices in data visualization, which advocate for simplicity and clarity, especially when conveying complex interactions in a multidimensional dataset.

Can you hypothesize why some species changed in the 4-weeks but not in the 8-weeks? Are the comparisons 4v0 and 8v0 or 4v0 and 8v4?

RESPONSE: As described at lines 193-194 and within Figure 1h-i, these comparisons were made over baseline (i.e., 4v0 and 8v0). The differences in microbial responses between the 4-week and 8-week time points are indeed intriguing and may reflect the complex dynamics of the gut microbiome in response to dietary alterations. Several factors could contribute to this phenomenon, and we added mention of probable community resilience at lines 199-204.

Community resilience is an interesting suggestion.

Reviewer #4 has suggested a possible reason for seeing results in the IF-P diet and not in the CR diet - if the run-in period is more similar to the CR diet than it is to the IF-P diet, it introduces a major bias to the results, and the comparison between them should at least in the text address that. In line with my comments on the subject.

As stated above, the run-in baseline diets were similar between groups, while both intervention arms were significantly different from the baseline run-in diets. Therefore, by study design, no bias was introduced, as stated by the reviewer.

Please mention how big of a change each diet constitutes relative to the baseline of its participants, and not just between the diets. Is it the same for both diets? If it is greater in the IF-P than in the CR diet, many of the conclusions should be taken with care, such as “a pronounced increase in Bray-Curtis dissimilarity was observed in the IF-P compared to the CR group” and “We observed differential abundance patterns at the family and genus-level in response to the IF-P but not the CR intervention” as these are seemingly the expected results. Could it also be the mode of food intake, shakes vs whole foods, the reason for the differential results?

RESPONSE: We have added the intra-individual median Bray-Curtis dissimilarities for IF-P and CR groups at weeks 4 and 8 and lines 190-192. In addition, although meal replacement shakes were used in IF-P, they consumed at least equal amounts of whole foods (>50%) throughout the 8-week intervention. Regardless, we agree with the reviewer that differences in the mode of calorie intake between groups may have influenced the results, as described in the manuscript (lines 353-55; 646-48). Despite the mode of calorie intake differing, liquid meal replacement products are popular among consumers for weight loss and weight maintenance plans. As such, this study design adds real-world application for popular dietary interventions.

I ask about diet, the response is about microbiome. This is a main point regarding the study design the authors do not address.

Please know, there were no differences in baseline run-in diets between groups for all dietary intake variables. Whereas, both intervention arm diets were significantly different from baseline (IF-P/CR vs. Baseline), as well as from each other (IF-P vs. CR).

“While the direction of influence wasn’t clear in this study (e.g., microbes influencing cytokine concentrations), the effect of inflammatory states is well-known to impact gut permeability.” Causality can not be determined in this study setting. The changes in cytokines and microbiome can be independent of each other, and both simply a direct result of the change in diet. In order to support the

claims mentioned, one should conduct a mediation analysis to show there is an indirect effect between the two factors .

RESPONSE: We agree that causality cannot be determined in the current study and have added this to the text, line 284.

If not conducting a mediation analysis, please acknowledge the microbiome and cytokines correlated effect can both be a direct result of the diet, in the results section and the discussion part pointing to these results.

We agree with acknowledging this limitation and have provided additions to the Results (see lines: 308-320) and Discussion sections (see lines: 706-709).

“100-fold leave-one-out cross-validation” please explain how you ended up with 100 samples at this stage.

RESPONSE: To clarify, the term "100-fold leave-one-out cross-validation" in our manuscript refers to the statistical validation technique we employed, not the number of samples. In this approach, each single sample from our dataset was left out in turn, and the model was trained on the remaining samples. This process was repeated 100 times to ensure robustness and to minimize bias, providing a comprehensive validation of our model. The number '100' here represents the iterations of cross-validation, not the number of samples.

K-fold is a term used for splitting the data to K equal size parts. Leave-one-out is when you take one sample out for a test set, and do it for all the samples. Please rephrase to make it clear it is 100 repetitions and not splitting the data to 100 equal size parts.

Thank you for your valuable feedback and for seeking further clarification regarding our validation technique. We acknowledge the importance of accurately describing our statistical methods to avoid any confusion. Our reference to "100-fold leave-one-out cross-validation" was indeed intended to describe the repetition of the leave-one-out cross-validation (LOOCV) process 100 times to enhance the robustness of our model validation. To clarify, in each iteration of LOOCV, one sample is left out as the test set, and the model is trained on the remaining samples. This process is not a division of data into 100 equal parts but rather a repeated application of LOOCV to ensure comprehensive evaluation and minimize bias. We appreciate your guidance on the matter and have revised the manuscript accordingly to ensure the description is precise and aligns with standard terminology. This has been added at lines 1043-1049 in the Methods section.

Both sections of the results regarding metabolites that are displayed in Figure 3, show greater change in the CR than in the IF-P diet. However, the text is focused on the IF-P group. The authors try to mitigate this point by saying “the greater number of affected pathways and metabolites in the CR group may have been related to the overall greater dietary diversity”. To the reader, it is unclear there is a dietary diversity difference between the groups, and in general this claim does not settle with the microbiome results that were more abundant in the IF-P group

RESPONSE: The reviewer raises a valid point. However, overall, the metabolite differences favored a greater reliance on fat oxidation and less sugar alcohol metabolism in IF-P compared to CR, which align with the significant improvements in GM and greater weight loss and body composition improvements

in IF-P study participants. These were the primary outcomes of interest. However, it's worth noting that CR had favorable changes in certain amino acid metabolites related to longevity and reduction in cancer risk that may or may not be related to dietary diversity. We have revised this in the text at lines 374-376.

To the reader, it is still unclear there is a dietary diversity difference between the groups.

See explanation of baseline run-in diets versus intervention diets above.

Please use scientific statistical language and methods, line 541 "visually", line 547 "notable", line 574 "observed". It is not clear why these results were described without scientific testing, while the results from line 576 onwards, were analyzed by a statistical method

RESPONSE: In this case study, we recognized the unique nature of the data, where traditional statistical tests were not applicable due to the extended and continuous duration of observations. As such, we employed a visual inspection method to identify notable patterns and trends. Moreover, we are currently analyzing all outcomes in this case study for a subsequent publication and will be highlighting the overall patterns and trends that may inform a larger and longer study.

"we provide evidence of long-term GM stabilization from these changes by following one individual over 12 months" as the case study is not statistically tested I would not call it "evidence".

We have changed "evidence" to "support".

"Pathway analysis identified significant impacts" the p-values are not corrected for multiple hypothesis testing so I would not call it "significant".

The word "significant" was removed.

The limitation paragraph in the discussion is good, please add other limitations that came up in this review such as the food intake mode (shakes vs solid food) that could affect among other things gut symptomatology, and the magnitude of dietary nutrients change which could affect the magnitude of change seen in each group. Another limitation is that dietary intake and adherence are self reported

"These findings highlight the importance of personalized approaches" can you explain how this study relates to personalization? the connection is unclear

RESPONSE: We are grateful for these additional limitations and have included them in the revised manuscript at lines 717-723. Regarding the findings impact on personalized approaches, a greater understanding of how different dietary regimens impact the microbiome may inform precision nutrition strategies moving forward, especially regarding body composition and cardiometabolic health lifestyle improvements.

The magnitude of dietary nutrients change which could affect the magnitude of change seen in each group is still not addressed.

Please see above response relating to dietary nutrients change.

Reviewer #3 (Remarks to the Author):

Previous concerns regarding interpretation of dietary fiber intake persist. The authors continue to use dietary fiber too comprehensively given the nature of the IF-P intervention. Fiber from whole foods exists in a complex matrix whereas fiber from isolated sources produces distinct effects on the microbiome. In the case of the IF-P arm, while grams of dietary fiber were, for all intents and purposes, similar to baseline and the CR arm, the introduction of 18g (70% of total IF-P arm fiber intake) being from an isolated fiber source would be expected to produce distinct effects (See PMID: 32846882 for limitations of the generic term "dietary fiber"). Given the nature of the intervention arms and study design, the IF-P arm vs CR arm was similar to an isolated fiber supplement study (e.g. PMID: 37444219). This needs to be given more consideration and appreciation when comparing the study arms and interpreting results.

We are grateful for Reviewer 3's helpful feedback and recommendations throughout the review process.

We agree with the reviewer and have updated the manuscript to reflect these valid concerns on lines 230-242. Specifically, we note that in addition to 30% greater fiber intake from IF-P versus CR, that the dietary fiber intake profile is likely different between groups (primarily resistant starch on IF-P versus a more complex, soluble/insoluble dietary fiber profile derived from vegetables and legumes consumed by those on CR). This differentiation is likely also influenced by the food consumption pattern between groups – for example, IF-P consumers of the meal replacement shakes would provide quick and immediate exposure of their fiber to the gastrointestinal tract versus CR followers providing their complex fiber profile slowly upon meal consumption. Collectively, these factors are likely contributing to overall gut microbial composition and structure, despite similarities in SCFA biosynthesis.

Notably, however, the effects of the IF-P intervention are still impressive and worth the continued effort the authors have put forth in their analysis.

We appreciate the reviewers support and understanding of the novelty and importance of the data.

Reviewer #4 (Remarks to the Author):

Thank you for the opportunity to review a revised manuscript entitled "Gut microbiome remodeling and metabolomic profile in response to protein pacing with intermittent fasting versus continuous caloric restriction" by Dr Mohr et al. The authors have made significant efforts to address reviewer comments.

We thank Reviewer 4 for their thoughtful feedback and constructive comments on our revised manuscript.

There are few minor comments:

Figure 1 shows most analysis where data in weeks 4 and 8 are compared to baseline (week 0) data except for Fig.1g (community composition) where the IF-P at weeks 4 and 8 is compared to CR and not a shift from baseline to the other weeks separately for IF-P and CR. Basically week 0 is missing. Can authors include time point 0 to Fig.1g to conform with the rest of the sub figures in Fig.1 or is there a reason why that was not included? Especially when authors refer to a shift from baseline to other time points in their results section in line 185-186 (“To understand the taxa driving this GM variation from baseline to weeks four and eight between the two dietary interventions...”) yet the data/figure is not represented.

To clarify, Fig. 1g is designed to illustrate intra-individual Bray-Curtis dissimilarities, focusing on the shifts within the gut microbiome composition from baseline to weeks four and eight for each dietary intervention group. As detailed in our methods section, the Bray-Curtis dissimilarity metric is employed to quantify the within-subject changes in microbial community composition by comparing paired samples (i.e., baseline vs. weeks four and eight) for each participant. This approach inherently involves comparing each individual's microbial composition at subsequent time points against their baseline, rather than presenting a direct, isolated comparison of the week 0 community composition. Since the aim is to demonstrate the extent of microbiome variation within individuals as they undergo dietary interventions, the baseline serves as the reference point for each subsequent time point comparison, rather than as a standalone condition.

We acknowledge, however, that this approach might differ from the presentation of data in other subfigures of Figure 1, where baseline comparisons are more explicitly depicted. Considering your helpful feedback, we adjusted the text accompanying Fig. 1g to ensure that our explanation of the analytical approach is clear. The caption now reads: “(g) Intra-individual changes in GM community structure from baseline to weeks four and eight in of IF-P participants shifted significantly throughout the IF-P intervention compared to CR as measured by the Bray-Curtis dissimilarity index (Wilcoxon rank-sum test).”

Fig5, ie line 507 (5j) legend reads: “Grid-fused least absolute shrinkage and selection operator (LASSO) regression” - is it GFLASSO or LASSO? Also check the legend for Figures2 in the supplemental information, it also has LASSO.

Thank you for pointing out this discrepancy. We have corrected this in the manuscript and supplemental file.

Line 664-665 in the discussion reads: “In contrast, the Low-responder group exhibited a decreased abundance of butyrate-producing bacteria.”

We apologize for this oversight. This was meant to read as “increased”. We have adjusted this sentence.

The results in Fig. 5e does not appear to suggest that, if anything at all, there is comparable enrichment in the abundance of butyrate producing bacteria for both low and high responders. Can authors explain how they arrive at that conclusion? since I find that text being somehow overstated.

It appears the line numbers in the STORMS checklist do not match with the main document (pdf version). Here are 3 examples under Methods section:

Study Design: Authors state line 95, I find that information in line 92.

Antibiotics usage: Authors state line 743, that information appears in a different line number.

Ethics: Authors state line 749-752, information is not available in the stated line numbers.

Can authors recheck the whole checklist to make sure all items conform to line numbers in the main text?

We have corrected this in the resubmission to ensure the STORMS checklist accurately corresponds to the line numbers in the main text of the manuscript without tracked changes. We apologize for any confusion this may have caused and are grateful for the opportunity to correct these inaccuracies.

Line 116 ANOVA's should be "ANOVA"

This has been corrected.

Delete the empty table after TableS1 in supplemental information

This has been corrected.

Reviewer #4 (Remarks on code availability):

The account has been created but the code is not available for review.

Thank you for bringing this to our attention. As previously mentioned, our initial plan was to make the scripts publicly available in the project repository "prior to publication," with the intention of incorporating any final revisions based on the review process. We recognize, however, the importance of having access to the code for a thorough review. To facilitate this, we have now uploaded the code to the project repository to aid in the review process. We hope this addresses your concern and assists in the evaluation of our manuscript.

REVIEWERS' COMMENTS

Reviewer #2 (Remarks to the Author):

My concerns were addressed, I apologize for previously missing figure S1b.

Regarding the blindness of arm assignment, even if it is an acceptable limitation, it could still be mentioned. Anyway, this is a minor comment.

Reviewer #3 (Remarks to the Author):

The authors did an excellent job addressing likely microbial substrate differences between diets. No additional concerns.

Reviewer #4 (Remarks to the Author):

The authors have addressed all comments except not being consistent with "GFLASSO". The article file/manuscript (e.g. line 551, figure 5j legend) and supplementary information (e.g. lines 46, 216) have a mix of "GFLASSO" and "GLASSO"

For a greater appreciation of the differences between the baseline run-in diet and the intervention diets (IF-P, CR), authors should kindly provide the dietary regimen for the run-in diet as a supplemental table similar to the table they provide for IF-P and CR diets in Table S1. I could not find the run-in dietary regimen information in reference# 15.

Reviewer #4 (Remarks on code availability):

The codes provided are reproducible

REVIEWERS' COMMENTS NCOMMS-23-43398C

Reviewer #2 (Remarks to the Author):

My concerns were addressed, I apologize for previously missing figure S1b.

Regarding the blindness of arm assignment, even if it is an acceptable limitation, it could still be mentioned. Anyway, this is a minor comment.

We are grateful for Reviewer 2's support and helpful feedback throughout the review process.

Reviewer #3 (Remarks to the Author):

The authors did an excellent job addressing likely microbial substrate differences between diets. No additional concerns.

Thank you for the excellent feedback and support of our novel work.

Reviewer #4 (Remarks to the Author):

The authors have addressed all comments except not being consistent with "GFLASSO". The article file/manuscript (e.g. line 551, figure 5j legend) and supplementary information (e.g. lines 46, 216) have a mix of "GFLASSO" and "GLASSO"

Thank you for the insightful and constructive comments during each review, the manuscript has been strengthened because of Reviewer #4's feedback. We have revised the manuscript as suggested by Reviewer #4 and only use "GFLASSO" throughout the manuscript.

For a greater appreciation of the differences between the baseline run-in diet and the intervention diets (IF-P, CR), authors should kindly provide the dietary regimen for the run-in diet as a supplemental table similar to the table they provide for IF-P and CR diets in Table S1. I could not find the run-in dietary regimen information in reference# 15.

We appreciate the Reviewer bringing this to our attention and have added reference to the comparison of the baseline run-in diets to the intervention diets at line 111. Please note, the differences between the baseline run-in diet and the intervention diets for all dietary intake variables is provided in Table 2 of reference #15.

Reviewer #4 (Remarks on code availability):

The codes provided are reproducible

Thank you.